

# LGM climate forcing and ocean dynamical feedback and their implications for estimating climate sensitivity

Jiang Zhu[1], Christopher J. Poulsen[2]

[1]Climate and Global Dynamics Laboratory, National Center for Atmospheric Research, Boulder, CO 80305, USA

[2]Department of Earth and Environmental Sciences, University of Michigan, Ann Arbor, MI 48109, USA

*Correspondence to*: Jiang Zhu (jiangzhu@ucar.edu)

**Abstract.** Equilibrium climate sensitivity (ECS) has been directly estimated using reconstructions of past climates that are different than today's. A challenge to this approach is that temperature proxies integrate over the timescales of the fast feedback processes (e.g. changes in water vapor, snow, and clouds) that are captured in ECS as well as the slower feedback processes

(e.g. changes in ice sheets and ocean circulation) that are not. A way around this issue is to treat the slow feedbacks as climate forcings and independently account for their impact on global temperature. Here we conduct a suite of Last Glacial Maximum (LGM) simulations using the Community Earth System Model version 1.2 (CESM1.2) to quantify the forcing and efficacy of land ice sheets (LIS) and greenhouse gases (GHG) in order to estimate ECS. Our forcing and efficacy quantification adopts the effective radiative forcing (ERF) and adjustment framework and provides a complete accounting for the radiative,

topographic, and dynamical impacts of LIS on surface temperatures. ERF and efficacy of LGM LIS are –3.2 W m$^{-2}$ and 1.1, respectively. The larger-than-unity efficacy is caused by the relatively larger temperature changes over land and the Northern Hemisphere subtropical oceans than those in response to a doubling of atmospheric $CO_2$. The subtropical SST response is linked to LIS-induced wind changes and feedbacks in ocean-atmosphere coupling and clouds. ERF and efficacy of LGM GHG are –2.8 W m$^{-2}$ and 0.9, respectively. The lower efficacy is primarily attributed to a smaller cloud feedback at colder

temperatures. Our simulations further demonstrate that the direct ECS calculation using the forcing, efficacy, and temperature response in CESM1.2 overestimates the true value in the model by 25% due to the neglect of slow ocean dynamical feedback. This is supported by the greater cooling (6.8°C) in a fully coupled LGM simulation than that (5.3°C) in a slab ocean model simulation with ocean dynamics disabled. The majority (67%) of the ocean dynamical feedback is attributed to dynamical changes in the Southern Ocean, where interactions between ocean stratification, heat transport, and sea-ice cover are found to

amplify the LGM cooling. Our study demonstrates the value of climate models in the quantification of climate forcings and the ocean dynamical feedback, which is necessary for an accurate direct ECS estimation.

## 1 Introduction

Equilibrium climate sensitivity (ECS) is defined as the global mean surface air temperature (GMST) response to a doubling of atmospheric $CO_2$ and accounts for the Planck response and water vapor, ice albedo, lapse rate, and cloud feedbacks (with



timescales < 100 years; Charney et al., 1979). Estimates of ECS range from 1.5–4.5°C and have remained nearly unchanged for the past 40 years (IPCC, 2013). Part of the reason for this large range is the overall weak constraints from present-day observations due to the brevity of instrumental record, the small magnitude of radiative forcing and temperature response relative to the natural variability, and the dependence of ECS estimates on the transient sea-surface temperature (SST) pattern in the historical period (Knutti, Rugenstein, & Hegerl, 2017).

Paleoclimate records overcome these limitations and provide unique observational constraints on ECS. The Last Glacial Maximum (LGM; ~21 ka BP) has been considered to be an ideal target for estimating ECS since it represents a quasi-equilibrium climate state with large changes in climate forcing and response and relatively high spatial coverage of well-dated proxy temperatures. Rohling et al. (2012) proposed a framework to obtain ECS from reconstructions of paleo-temperatures and climatic forcings in which slow (timescales > 100 years) feedback processes such as changes in GHGs, LISs, Earth's

orbits, and land use are considered as climate forcings rather than feedbacks. This approach has been widely used to directly calculate ECS from proxy reconstructions of the LGM and glacial-interglacial cycles with estimates ranging from 2.6 to 8.1°C (Friedrich, Timmermann, Tigchelaar, Elison Timm, & Ganopolski, 2016; Köhler et al., 2017; Stap, Köhler, & Lohmann, 2019; Tierney, Zhu, et al., 2019; von der Heydt, Köhler, van de Wal, & Dijkstra, 2014).

Direct ECS estimation using the Rohling et al. (2012) approach relies on a complete understanding of the slow feedback

processes. In the context of the LGM, both the radiative forcing due to changes in GHGs, LISs, vegetation, and aerosols and their efficacy (impact on GMST) must be known. Of these forcings, GHGs and LISs have been considered in most LGM-based ECS estimations (Friedrich et al., 2016; Köhler et al., 2017; Schmittner et al., 2011; Stap et al., 2019; Tierney, Zhu, et al., 2019; von der Heydt et al., 2014). Previous estimates of the LGM LIS forcing account for albedo changes associated with the presence of LISs and the exposure of shelves due to the lowered sea level (Figure 1a), yielding a shortwave forcing ranging

from –1.5 to –5.2 W m$^{-2}$ (Braconnot et al., 2012; Braconnot & Kageyama, 2015; Friedrich et al., 2016; Hansen, Sato, Russell, & Kharecha, 2013; Köhler et al., 2010; Taylor et al., 2007; Tierney, Zhu, et al., 2019). However, these estimates neglect changes in surface topography (Figure 1b), which can change surface temperature and longwave emission. Moreover, LIS topographic changes altered atmospheric (Figure 1c) and ocean circulations (Herrington & Poulsen, 2012; Kutzbach & Guetter, 1986; Zhu, Liu, Zhang, Eisenman, & Liu, 2014), which can change surface temperatures without directly involving radiative

processes (e.g., through a wind-evaporation-SST feedback (Xie & Philander, 1994)). To our knowledge, a complete quantification of the LGM LIS forcing that accounts for the radiative, topographic, and dynamical effects has not been done. Furthermore, the efficacy of LIS forcing has received relatively little attention (Yoshimori, Yokohata, & Abe-Ouchi, 2009), although it is clear that albedo effects are mostly distributed over high latitude land. This oversight is problematic since assumptions of LIS efficacy can greatly impact the resulting ECS estimates (Stap et al., 2019; Tierney, Zhu, et al., 2019).

Another caveat with the direct ECS estimation approach is that fast feedbacks depend on the climate state and interact with slow feedbacks. Global climate model (GCM) studies have shown that the magnitude of fast climate feedbacks vary with global temperature (e.g., stronger cloud and water vapor feedbacks at higher GMSTs; (Caballero & Huber, 2013; Schneider,





Kaul, & Pressel, 2019; Yoshimori et al., 2009; Zhu & Poulsen, 2020; Zhu, Poulsen, & Tierney, 2019). Furthermore, the LGM ocean circulation was different than today (e.g., Curry & Oppo, 2005). Ocean dynamical processes can influence global

temperature through interactions with sea ice, SST pattern, and cloud processes (Dong, Proistosescu, Armour, & Battisti, 2019; Ferrari et al., 2014; Rose, Armour, Battisti, Feldl, & Koll, 2014; Shin et al., 2003; Winton, Griffies, Samuels, Sarmiento, & Frölicher, 2013; Zhou, Zelinka, & Klein, 2017), constituting an ocean dynamical feedback that takes place on timescales longer than 100 years. The contribution of the ocean dynamical feedback to the magnitude of LGM cooling and its impact on direct ECS estimation have not been thoroughly studied.

In this study, we address whether ECS can be accurately estimated using the direct calculation approach and knowledge of the LGM climate forcing and global temperature. To answer this question, we adopt the adjusted forcing-feedback framework (Sherwood et al., 2014) to provide a complete quantification of the forcing and efficacy of LGM LIS and GHG using a suite of climate simulations. In contrast to previous studies that only considered surface albedo effects, our analysis of LIS forcing and efficacy accounts for LIS radiative, topographic, and dynamical effects. We also investigate the role of the ocean dynamical

feedback in modulating the magnitude and spatial distribution of the LGM cooling by comparing fully coupled and slab ocean simulations. Finally, we discuss the implications of our results for direct ECS estimation using paleoclimate reconstructions.

## 2 Method, model, and experiments

### 2.1 Model and fully coupled simulations

We employ the Community Earth System Model (CESM) version 1.2 with a horizontal resolution of 1.9 × 2.5° (latitude ×
longitude) for the atmosphere and land, and a nominal 1° for the sea ice and ocean (Hurrell et al., 2013). CESM1.2 is among the models that best reproduce climate features from instrumental records (Knutti, Masson, & Gettelman, 2013) and has been extensively used for studying past climates (DiNezio et al., 2016; Feng et al., 2019; Otto-Bliesner et al., 2015; Tierney, Haywood, Feng, Bhattacharya, & Otto-Bliesner, 2019; Zhu, Liu, Brady, Otto-Bliesner, Zhang, et al., 2017; Zhu, Liu, Brady, Otto-Bliesner, Marcott, et al., 2017; Zhu, Poulsen, & Otto-Bliesner, 2020; Zhu et al., 2019). Our CESM1.2 experiments were

run with prescribed satellite phenology (SP) in the land model (Community Land Model version 4; CLM4) without an active carbon-nitrogen (CN) biogeochemical cycle. In the SP mode, leaf area and stem area indices, and vegetation heights in CLM4 are prescribed according to data derived from satellite observations (Lawrence et al., 2011). Our choice of CLM4 with satellite phenology is based on the overall poorer simulation of vegetation phenology with an active CN, which could potentially be more problematic for paleoclimate simulations (Lawrence et al., 2011).

We make use of the fully coupled preindustrial and LGM simulations (FCM_PI and FCM_LGM; Table 1) from our previous studies (Tierney, Zhu, et al., 2019; Zhu, Liu, Brady, Otto-Bliesner, Zhang, et al., 2017). FCM_LGM was forced by boundary conditions consistent with protocols from the Paleoclimate Modelling Intercomparison Project phase 4 (PMIP4), including altered GHG concentrations, Earth orbital parameters, and LIS (Kageyama et al., 2017). FCM_LGM used prescribed





preindustrial vegetation cover and aerosol emissions, as reliable global reconstructions are not available (Kageyama et al.,
2017; Köhler et al., 2010). Although FCM_LGM contains the orbital forcing, its effect on GMST is small (Liu et al., 2014)
and neglected in the following analysis. The FCM_LGM ocean state was initialized from the LGM simulation using the
Community Climate System Model version 4, which had been spun-up for more than 2,400 years and reached a quasi-
equilibrated state (Brady, Otto-Bliesner, Kay, & Rosenbloom, 2013). FCM_LGM was integrated for an additional ~1,800
years to reach equilibration under a different atmosphere model (DiNezio et al., 2016; Tierney, Zhu, et al., 2019; Zhu, Liu,
Brady, Otto-Bliesner, Zhang, et al., 2017). The TOA energy imbalance averaged over the last 100 years is –0.06 W m$^{-2}$ in
FCM_LGM, which is comparable to the 0.09 W m$^{-2}$ in FCM_PI, indicating the surface climate has reached a quasi-equilibrium
glacial state. GMST in FCM_LGM is 6.8°C lower than that in FCM_PI and in agreement with emerging evidence for an LGM
cooling of approximately 6°C (Tierney, Zhu, et al., 2019).

## 2.2 "Fixed SST" simulations and the effective radiative forcing

We adopt the forcing-feedback framework with the concept of rapid adjustments (Sherwood et al., 2014). We use fixed-SST
experiments to calculate the effective radiative forcing (ERF), defined as the change in net top-of-atmosphere (TOA) radiative
flux after adjustments of the atmospheric temperature profile, water vapor, and clouds (Hansen et al., 2005). Our results show
that this method is especially well-suited for quantifying the LIS forcing and is an advancement over either simplified bulk
calculations or the approximate partial radiative perturbation method used in previous studies, which only provide an
estimation of the shortwave forcing from albedo effects. In the fixed-SST experiments, an LGM climate forcing (e.g., GHGs)
is introduced into a preindustrial simulation with active atmosphere and land models, but with SST and sea ice prescribed to
the unperturbed preindustrial climatology. The land surface temperature is allowed to adjust as it is impractical to fix in the
model. The ERF attributed to a forcing is obtained as the change in TOA net radiation between simulations with and without
the forcing. ERF and land temperature change in the fixed-SST experiments are termed ERF$_{fsst}$ and $\Delta T_{fsst}$, respectively.
Changes in atmospheric temperature, water vapor, and clouds in response to the climate forcing, without mediation by the
global-mean temperature, are referred to as "adjustments".

To quantify the ERF due to LGM LIS and GHG, we performed two fixed-SST experiments with LGM GHG and LIS forcing,
respectively (ATM_GHG and ATM_LIS; Table 1). To examine whether ERFs from GHG and LIS are additive, we performed
an additional experiment with both forcings included (ATM_LGM). To compare the forcing and response of LGM GHG and
LIS to CO$_2$ increasing, we also carried out an experiment with twice the pre-industrial atmospheric CO$_2$ concentration
(ATM_2CO2). Finally, a standard preindustrial atmosphere-only simulation was performed and used as a reference (ATM_PI).
These fixed-SST experiments used the same set of preindustrial SST and sea ice coverage derived from the FCM_PI
climatology. All simulations were run for 30 years with the last 25 used for analysis.

ERF$_{fsst}$ contains radiation changes resulting from land-surface temperature changes ($\Delta T_{fsst}$), which bias ERF$_{fsst}$. For example,
$\Delta T_{fsst}$ is negative in ATM_GHG (see Figure 2a), leading to an underestimation of the magnitude of ERF due to the decrease




of Planck emission at lower land surface temperatures. To account for the radiative effects from $\Delta T_{fsst}$, two corrected versions of ERF were computed. In the first, the $\Delta T_{fsst}$ effect on TOA radiation was corrected using the climate sensitivity parameter ($\lambda$; units: K W$^{-1}$ m$^{-2}$) as:

$$ERF_\lambda = ERF_{fsst} - \Delta T_{fsst} / \lambda \qquad (1).$$

$\lambda$ was obtained from the coupled simulation in a slab ocean configuration (See Section 2.3). In the second correction, radiative kernels were used:

$$ERF_{kernel} = ERF_{fsst} - A_{Ts} - A_{Ta} - A_q - A_{alb} \qquad (2).$$

In this approach, $ERF_{kernel}$ is obtained by subtracting the direct rapid adjustments associated with $\Delta T_{fsst}$ from $ERF_{fsst}$ while keeping the indirect rapid adjustments, such as cloud responses (Tang et al., 2019). The direct rapid adjustments that are

subtracted include effects over land from changes in surface temperature ($A_{Ts}$), tropospheric air temperature ($A_{Ta}$), tropospheric water vapor ($A_q$), and albedo ($A_{alb}$). $A_{Ta}$ is calculated by assuming a constant lapse rate in the troposphere, i.e. the same tropospheric air temperature change as the surface. Similarly, $A_q$ accounts for the effect from tropospheric water vapor change under the assumption of a constant lapse rate. $A_q$ is calculated by scaling the total water vapor effect with the ratio between the temperature-induced radiative flux change from a constant lapse rate and that from the full tropospheric

temperature change. In the calculation, we employed the radiative kernels that were specifically developed for CESM (Pendergrass, Conley, & Vitt, 2018). We note that radiative kernels developed for the present-day climate could not be applied to our ATM_LIS case, due to the drastically different topography and land surface properties.

**2.3 SOM simulations and the efficacy of forcing**

To compare temperature responses to different climate forcings and to estimate $\lambda$, we performed sensitivity experiments in a

slab ocean model (SOM) configuration (Table 1) without ocean dynamics. The SOM uses prescribed mixed layer depth and heat transport convergence ("q-flux" hereafter) (Bitz et al., 2011) that are derived from FCM_PI. As a result, temperature responses in SOM simulations are caused by the fast feedback processes and exclude the ocean dynamical feedback. SOM_GHG includes LGM GHG levels and the preindustrial values of all the other boundary conditions. Similarly, SOM_LIS incorporates the LGM ice-sheet forcing including a higher topography, an altered land-sea distribution to account for effects

from sea-level change, and modified land surface properties over ice sheets. To examine whether the climate responses are additive, we performed SOM_LGM, in which both LGM GHG and LIS forcings were added. In addition, we conducted SOM_2CO2 with $CO_2$ level two times the preindustrial value. These SOM simulations were integrated for 60 years to allow the model to reach equilibrium (with TOA energy imbalance < 0.1 W m$^{-2}$). Averages over the last 20 years are used for analysis.

The climate sensitivity parameter ($\lambda$) was obtained from fixed-SST and SOM simulations and used to calculate $ERF_\lambda$ (Equation (1)). Specifically, for a climate forcing (GHG, LIS, or 2CO2), $\lambda$ was estimated as:



$$\lambda = \frac{\Delta T_{SOM} - \Delta T_{fsst}}{ERF_{fsst}} \qquad (3).$$

$\Delta T_{SOM}$ is the equilibrated GMST response in the SOM simulation. $\Delta T_{SOM} - \Delta T_{fsst}$ represents the SST-mediated surface air temperature changes that is associated with $ERF_{fsst}$.

We define the efficacy ($\varepsilon$) of a climate forcing (GHG or LIS) as a ratio of its temperature response scaled by its $ERF_\lambda$ to that of 2CO2:

$$\varepsilon = \frac{\Delta T_{SOM}/ERF_\lambda}{(\Delta T_{SOM}/ERF_\lambda)_{2CO2}} \qquad (4).$$

## 2.4 The radiative kernels and APRP approach

To correct the ERFs of doubling $CO_2$ and LGM GHG and to understand their efficacy, we employ the radiative kernels that
are developed for CESM (Pendergrass et al., 2018). In the analysis, we calculate changes in the 12-month climatology of variable of interest (e.g., surface temperature) and multiply that by the corresponding radiative kernel to estimate the TOA radiation changes. Climate feedback parameters are obtained by normalizing the TOA radiation anomalies by the GMST changes. Kernels analyses are not performed for LGM LIS simulations, as the present-day kernels are not suitable due to the large difference in the characteristics of the forcing and response (Yoshimori, Hargreaves, Annan, Yokohata, & Abe-Ouchi,
170    2011).

The approximate partial radiative perturbation (APRP; Taylor et al., 2007) is used to quantify the shortwave forcing and feedback, in particular for the LIS simulations. In contrast to the radiative kernels method, APRP is independent of the forcing and background climate state and produces results that differ from PRP by less than 7% (Taylor et al., 2007; Yoshimori et al., 2011; Zhu & Poulsen, 2020; Zhu et al., 2019). APRP represents the atmosphere as a single layer with bulk optical properties
and usually uses monthly mean model output to derive the radiative effects and feedbacks associated with changes in surface albedo, clear-sky processes, and clouds. The shortwave cloud feedback is further decomposed into contributions from changes in cloud amount, scattering, and absorption. APRP has been used in many previous studies to quantify the shortwave forcing associated with LIS albedo changes (Braconnot et al., 2012; Braconnot & Kageyama, 2015; Brady et al., 2013; Taylor et al., 2007; Tierney, Zhu, et al., 2019). When using APRP to quantify shortwave feedbacks, we only show results over model grid
points that are ocean in both PI and LGM simulations; in this way, climate feedbacks are separated from forcings, e.g. from LIS, although feedback processes over land are overlooked.



## 3 Results

### 3.1 Effective radiative forcing

The $ERF_{fsst}$ due to LGM GHG is $-2.6\pm0.2$ W m$^{-2}$ ($\pm1\sigma$; Table 1; Figure 2c,e). After correcting the radiative effects associated

with land temperature changes ($\Delta T_{fsst} = 0.2°C$; Figure 2a) using the climate sensitivity parameter and radiative kernels, the $ERF_{\lambda}$ and $ERF_{kernel}$ are $-2.8\pm0.3$ and $-2.8$ W m$^{-2}$, respectively, and agree well with previous estimates of $-2.8$—$3.0$ W m$^{-2}$ (Hansen et al., 2013; Köhler et al., 2010). For a doubling of $CO_2$, $ERF_{fsst}$, $ERF_{\lambda}$, and $ERF_{kernel}$ are $3.7\pm0.3$, $3.9\pm0.3$, and $4.0$ W m$^{-2}$, respectively, well within the multi-model range in recent studies (Smith et al., 2018; Tang et al., 2019). For both LGM GHG and 2CO2, $ERF_{kernel}$ falls in the middle of the uncertainty range of $ERF_{\lambda}$, suggesting that both the correction methods

using radiative kernels and climate sensitivity parameters produce meaningful and accurate results.

In response to LGM LIS, $\Delta T_{fsst}$ has a global mean of $-1.3$ °C with maximum cooling over ice sheets exceeding 24°C, much greater than the land temperature changes associated with GHG forcing (Table 1; Figure 2a,b). $\Delta T_{fsst}$ results from the higher surface albedo over regions with increased coverage of ice sheets and land, the elevated ice-sheet topography, and radiative and dynamic atmospheric adjustments. The global mean $ERF_{fsst}$ due to LGM LIS is $-1.9\pm0.2$ W m$^{-2}$, resulting from a

shortwave component of $-3.7$ W m$^{-2}$ ($ERF_{fsst\_sw}$; Figure 2d) and a longwave component of $1.8$ W m$^{-2}$ ($ERF_{fsst\_lw}$; Figure 2f). $ERF_{fsst\_sw}$ is lowest over ice sheets with values less than $-80$ W m$^{-2}$. Using shortwave APRP, we attribute 77% ($-2.8$ W m$^{-2}$) of $ERF_{fsst\_sw}$ to surface albedo changes over regions of ice sheets and shelf exposure, 13% ($-0.5$ W m$^{-2}$) to surface albedo changes associated with snow cover increases outside the ice sheet regions, and 8% ($-0.3$ W m$^{-2}$) to cloud adjustments. The majority of the cloud adjustments ($-0.2$ W m$^{-2}$) occurs over the Indo-Pacific warm pool, where the exposure of the Sunda and

Sahul shelves produces a surface cooling and drying (DiNezio et al., 2016) and increases cloud condensates through enhanced large-scale moist advection (Zhang, Lin, Bretherton, Hack, & Rasch, 2003). Outside the tropics, clouds diminish over ice sheets and in the downwind regions and shift with the position of the storm tracks; yet, the overall impact on the global mean $ERF_{fsst\_sw}$ is small. $ERF_{fsst\_lw}$ exceeds 40 W m$^{-2}$ over ice sheets, which results primarily from the reduced longwave radiation due to a higher effective emission elevation and lower temperatures (Figures 1b and 2b).

Using the climate sensitivity parameter (diagnosed in SOM_LIS; Table 1), we calculate a global mean $ERF_{\lambda}$ from LGM LIS of $-3.2\pm0.2$ W m$^{-2}$. Our calculation accounts for the radiative, topographic, and dynamic adjustments associated with LIS, in contrast to only the albedo effects considered in previous studies (Braconnot et al., 2012; Braconnot & Kageyama, 2015; Taylor et al., 2007; Tierney, Zhu, et al., 2019). Using the APRP approach in these previous studies, we calculate a shortwave forcing of $-2.8$ W m$^{-2}$ from surface albedo changes over regions with new ice-sheet coverage and shelf exposures in ATM_LIS.

We note that this APRP approach overestimates the LIS shortwave radiative forcing by including the radiative effect of snow increases over ice sheets (or regions with shelf exposure), as albedo of fresh snow is considerably larger than that of glaciers (0.8–0.9 versus ~0.6) (Cuffey & Paterson, 2010). The LIS-induced cooling increases the proportion of snow relative to rain, which reflects more shortwave radiation than that from ice-sheet albedo alone. The snow-induced overestimation is larger if



the cooling over ice sheets is greater. For example, the shortwave forcing from APRP analysis is greater in coupled simulation

than that in atmosphere-only simulation with fixed PI SST (e.g., –3.3 in FCM_LGM versus –2.8 W m$^{-2}$ in ATM_LIS), due to the greater cooling over ice sheets.

## 3.2 Efficacy of LGM GHG and LIS forcings

Our results suggest that the efficiency of lowering GHGs to LGM levels is smaller than that of doubling atmospheric $CO_2$ under PI conditions, i.e. the LGM GHG forcing has a smaller-than-unity efficacy of 0.9±0.1 (Table 1; Equation 4). In

SOM_2CO2, GMST increases by 3.6 K in response to an ERF$_\lambda$ of 3.9 W m$^{-2}$. In SOM_GHG, GMST decreases by 2.2 K in response to an ERF$_\lambda$ of –2.8 W m$^{-2}$. The lower $\varepsilon$ of LGM GHG forcing is caused by a weaker cloud feedback in response to cooling (Table 2). Using radiative kernels, we find that the Planck, albedo and combined lapse rate and water vapor feedbacks stay largely unchanged; however, the cloud feedback parameter is 30% smaller in SOM_GHG than in SOM_2CO2 (0.32 versus 0.46 W m$^{-2}$ K$^{-1}$). The decrease in the cloud feedback is due to the shortwave component; the cloud scattering feedback

is weaker in response to cooling than that to warming over high-latitude regions, leading to a weaker shortwave response (Figure 3c,f; Table 2 APRP columns). These results demonstrate a state-dependent cloud feedback that increases with GMST, a feature that has been found in the latest three CAM versions (Zhu & Poulsen, 2020; Zhu et al., 2019).

The $\varepsilon$ of LGM LIS is 1.1±0.1, resulting from an ERF$_\lambda$ of –3.2 W m$^{-2}$ and a ΔGMST of –3.2 K in SOM_LIS (Table 1) and suggesting that LIS forcing is 10% more effective in changing GMST than doubling $CO_2$ when only fast feedbacks are

considered. 40% of the LIS-induced cooling (1.3 of 3.2 K) is attributed to land temperature changes that involve radiative, topographic, and dynamical effects of the LIS forcing and are independent from SST changes (Figure 1,2; Table 1); land only accounts for 7–8% of the GMST change in SOM_2CO2 and SOM_GHG. In addition to the large contribution from processes over land, the shortwave cloud feedback over ocean is greater in response to the LIS forcing (0.30 W m$^{-2}$ K$^{-1}$) than that to the doubling $CO_2$ forcing (0.31 versus 0.21 W m$^{-2}$ K$^{-1}$; APRP analysis in Table 2 and Figure 3). The greater shortwave cloud

feedback is due to changes in both cloud amount and scattering and is especially prominent over the Northern Hemisphere subtropics, which can be understood as an "SST pattern effect". The cloud feedback parameter is expressed as

$$\lambda_{CLD} \equiv \frac{d\text{CRE}}{d\text{GMST}} = \frac{\partial \text{CRE}}{\partial \text{SST}_{SUB}} \frac{d\text{SST}_{SUB}}{d\text{GMST}} \qquad (5),$$

where SST$_{SUB}$ is the SST over the subtropical North Pacific and North Atlantic, and CRE denotes the global cloud radiative effects. SST$_{SUB}$ is positively correlated with global CRE ($\frac{\partial \text{CRE}}{\partial \text{SST}_{SUB}} > 0$) (Dong et al., 2019; Zhou et al., 2017). As a result, a

greater SST$_{SUB}$ change relative to the GMST gives rise to a greater cloud feedback through changing the lower tropospheric stability (Wood & Bretherton, 2006). Over the subtropical North Pacific and North Atlantic, the shortwave cloud feedback exceeds 10 W m$^{-2}$ K$^{-1}$ in SOM_LIS, in comparison to a maximum of ~3 W m$^{-2}$ K$^{-1}$ in SOM_2CO2 (Figure 3c,f), which is consistent with the relative magnitude of SST change in each experiment (Figure 3b,h).



The formation of the SST responses over the subtropical North Pacific and North Atlantic is attributed to the ice sheet-driven

wind changes (Figure 3g). In response to the topographic effects of LGM LIS, the Northern Hemisphere westerly jet shifts

southward in ATM_LIS, producing cyclonic low-level wind anomalies over the subtropical and mid-latitude North Pacific and

anti-cyclonic anomalies over the subtropical North Atlantic (Kutzbach & Guetter, 1986; Zhu et al., 2014). This anomalous

wind pattern force regional SST changes through changing latent heat flux and amplify the coupled response through the wind-

evaporation-SST feedback (Chiang & Bitz, 2005; Xie & Philander, 1994). For example, the trade wind strengthens over the

subtropical North Atlantic, which cools the subtropical SST due to the enhanced evaporation and reinforces the anomalous

wind pattern, forming a positive feedback. This non-radiative pathway of LIS's influence on the surface temperature is largely

absent when GHGs are changed (Figure 3a,d), highlighting the complex nature of non-GHG climate forcings and the important

of using efficacy to evaluate the overall effectiveness of their radiative forcing.

### 3.3 Are forcing/responses additive?

Our simulations suggest that ERFs and surface temperature responses of LGM GHG and LIS are globally additive. In fixed-

SST experiments with both the LGM GHG and LIS forcings (ATM_LGM), $ERF_{fsst}$ is $-4.4\pm0.3$ W m$^{-2}$, approximately the

sum of those in ATM_GHG and ATM_LIS ($-2.6\pm0.2$ and $-1.9\pm0.2$, respectively). Similarly, the kernel-corrected $ERF_{\lambda}$ in

ATM_LGM, $-6.1\pm0.3$ W m$^{-2}$, is nearly equal to the sum of those in ATM_GHG and ATM_LIS ($-2.8\pm0.3$ and $-3.2\pm0.2$ W m$^{-2}$, respectively). The SOM_LGM $\Delta$GMST in response to combined GHG and LIS forcings is $-5.3\pm0.09°C$ and is close to the

sum of $\Delta$GMSTs in SOM_GHG and SOM_LIS ($-2.2\pm0.11$ and $-3.2\pm0.09°C$, respectively). From these results, we conclude

that ERF and $\Delta$GMST due to individual forcings are additive at the global level, which supports the approach to separate the

LGM climate forcing and response into components associated with individual forcing agents. We note that at the regional

level, especially over high latitudes, the ERF and $\Delta$GMST from the sum of individual forcings and combined forcings do not

match as well as at the global level (figure not shown), likely due to local feedbacks related to sea ice.

### 3.4 The ocean dynamical feedback

Our results demonstrate that the full extent of LGM cooling cannot be produced using a SOM configuration that accounts for

fast feedback processes but excludes the slow ocean dynamical changes. $\Delta$GMST is $-6.8°C$ in FCM_LGM and $-5.3°C$ in

SOM_LGM (Table 1; Figure 4a,b). Both simulations have reached an equilibrium state under the same climate forcings and

only differ in the complexity of the ocean model. The LGM cooling is $1.5°C$ (28%) greater with active ocean dynamics and

interactions with the atmosphere and sea ice. The difference in LGM cooling between SOM and FCM primarily occurs in the

Southern Ocean (SO), where SOM_LGM simulates a weaker LGM cooling by more than 10°C (Figure 4a–c). In the eastern

equatorial Pacific and eastern subtropical oceans (except for the subtropical Atlantic), SOM_LGM simulates a smaller LGM

cooling by 1–2°C. In the North Atlantic, LGM cooling in SOM_LGM is greater by ~1°C in the subtropics and more than 5°C

along the sea-ice margin in subpolar regions. Over the Indo-Pacific Warm Pool and the western subtropics, surface temperature





change is similar between FCM and SOM simulations, suggesting a limited role of ocean dynamical response over these regions.

Accounting for the SO dynamical effects increases the LGM cooling in the SOM configuration and explains the majority (67%) of the difference from FCM simulations. This is shown in an LGM SOM simulation (SOM_SO), in which the prescribed "q-flux" over the SO (90–40°S) is replaced with those derived from FCM_LGM, with other regions remaining unchanged 280 (using values from FCM_PI). SOM_SO simulates a colder LGM than SOM_LGM, especially over the SO, where the large temperature difference (>10°C) between SOM_LGM and FCM_LGM is mostly removed (Figure 4d versus c). In addition to the impact on local temperatures, SOM_SO simulates lower surface temperatures over the eastern equatorial Pacific and Indian Ocean and the Southern Hemisphere subtropics, producing a better match with FCM_LGM over these regions. This remote impact of the SO on low-latitude temperatures reflects displacement in tropical atmospheric circulations (Hwang, Xie, Deser, 285 & Kang, 2017), which is consistent with the zonal mean energetic theories (Kang, Held, Frierson, & Zhao, 2008; Schneider, Bischoff, & Haug, 2014).

The SO dynamical effects primarily result from ocean stratification changes and the coupling with sea ice. In FCM_PI, the SO is stratified with the maximum ocean temperature occurring in the subsurface (500–1000m; Figure 5a). In zonal and annual mean, isotherms (shadings in Figure 5a) intersect isopycnals (contours) mostly near 65–60°S and along the Antarctic coast, 290 indicating strong heat diffusion towards the mixed layer, i.e. a heat flux convergence of approximately –20 W m$^{-2}$ (Figure 5c; red curve). The strong heat flux convergence warms the mixed layer and inhibits sea-ice formation, resulting in a quasi-permanent sea-ice extent to 68°S (red horizontal bar in Figure 5c; defined using a 70% annual mean sea-ice cover). In comparison, SO stratification in FCM_LGM is greatly reduced with a potential density change of less than 0.4 kg m$^{-3}$ over most water columns and a largely invariant ocean temperature of –2°C (Figure 5b). Meanwhile, the centre of mixed-layer heat 295 flux convergence is shifted northward to ~56°S and the quasi-permanent sea ice expands to ~58°S (Figure 5c). These changes reflect a tight coupling between sea-ice extent and ocean dynamics in the SO (Ferrari et al., 2014; Shin et al., 2003). An initial LGM cooling in the SO (e.g., caused by fast feedback processes) increases sea-ice formation and brine rejection, which enhances convection and decreases stratification, resulting in a decrease of heat flux convergence to the mixed layer and amplifying sea-ice expansion. This feedback loop is absent in a SOM configuration with prescribed mixed layer depth and 300 heat flux convergence, leading to little expansion of the quasi-permanent sea ice (cyan horizontal bar in Figure 5c) and much less LGM cooling in SOM_LGM (Figure 4b,c). When the LGM changes in mixed layer depth and heat flux convergence is prescribed in SOM_SO, sea-ice expands northward (light green bar in Figure 5c) and the SO experiences cooling (Figure 4d versus c).

Accounting for additional ocean dynamical effects in the low latitudes and the Northern Hemisphere further decreases the 305 difference in LGM cooling between SOM and FCM simulations. This is supported by additional SOM simulations, in which we replace the prescribed "q-flux" with those derived from FCM_LGM over 90°S–30°N and the entire global ocean, respectively. The tropical ocean dynamical effects decrease SST in the eastern equatorial Pacific and the Southern Hemisphere





subtropics (Figure 4e). Ocean dynamics in the Northern Hemisphere middle and high latitudes increases SST over the North Atlantic (Figure 4f). The tropical ocean dynamical effects primarily result from changes in tropical ocean circulations and the coupling with the atmosphere (DiNezio et al., 2011; Vecchi & Soden, 2007). The Northern Hemisphere ocean dynamical effects are related to a stronger AMOC and a greater northward ocean heat transport (Brady et al., 2013). After accounting for the global ocean dynamical effects by using the "q-flux" derived from FCM_LGM, high-latitude oceans still exhibit a temperature difference of ~1–2°C, contributing to a GMST of ~0.3°C, likely reflecting the challenge of using prescribed "q-flux" to approximate the full extent of ocean dynamical effects (Bitz et al., 2011). Nevertheless, these results highlight the important role of the slow ocean dynamical feedback in modulating regional and global temperatures.

## 4 Discussion: implications for estimating climate sensitivity

Results presented herein highlight major caveats of the direct ECS estimation approach. Firstly, a complete understanding of the magnitude and efficacy of forcing agents is necessary, especially for non-GHG forcings (e.g. LIS, vegetation, and aerosols) that may have distinct spatial distribution and non-radiative pathways to change the energy balance of Earth. We suggest a GCM-based approach using the effective radiative forcing and adjustment framework to account for the complicated aspects of paleoclimate non-GHG forcings. In this approach, a fixed-SST simulation of ~30 years with a forcing of interest is first conducted to calculate the effective radiative forcing ($ERF_{fsst}$) and the associated land temperature changes. An $ERF_\lambda$ is then obtained by correcting the $ERF_{fsst}$ using the climate sensitivity parameter that is derived in an additional SOM simulation of ~60 years. Moreover, efficacy of the forcing can also be derived using these simulations. This approach provides a complete consideration of the radiative and non-radiative effects of the forcing agent and is more consistent with the basic definition of the forcing-feedback framework. In contrast, the APRP-based approach used in previous studies only accounts for the effects from albedo changes. We note that, due to the inclusion of snow effects in the forcing quantification, the APRP-based approach overestimates the shortwave albedo effects.

A second caveat concerns the role of the ocean dynamical feedback, which occurs on timescales of $10^2$–$10^3$ years and should be accounted for when directly estimating ECS using forcing/response of an equilibrium climate. This complication stems from defining ECS to include only fast feedback processes with timescales less than 100 years. Ocean feedback processes, including the heat redistribution by ocean circulation and the coupling with the atmosphere or sea ice, require more than 100 years to develop. Reconstructions of past climate forcings and GMST usually do not directly constrain ocean circulations and therefore could potentially impact the ECS estimation. We speculate that the ocean dynamical processes and their coupling with the atmosphere and sea ice may differ among climate models; this difference could explain the lack of correlation between global and regional mean LGM cooling and ECS in Paleoclimate Modelling Intercomparison Project models (Hargreaves, Annan, Yoshimori, & Abe-Ouchi, 2012; Hopcroft & Valdes, 2015).





To demonstrate the above caveats, we assume CESM1.2 is a perfect model and estimate ECS using LGM constraints that are derived from model simulations as:


$$ECS = \frac{\Delta GMST - \Delta GMST_{ODF}}{\varepsilon_{GHG} \times ERF_{GHG} + \varepsilon_{LIS} \times ERF_{LIS}} ERF_{2CO2} \qquad (6).$$

In the above equation, $\Delta GMST_{ODF}$ denotes the GMST change (approximately –1.5°C; see Section 3.4) that is caused by the slow ocean dynamical feedback and is subtracted from the total LGM cooling ($\Delta GMST$ = –6.8°C in CESM1.2). $ERF_{GHG}$, $ERF_{LIS}$, and $ERF_{2CO2}$ in our simulations are –2.8±0.3, –3.2±0.2, and 3.9±0.3 W m$^{-2}$, respectively (Table 1). $\varepsilon_{GHG}$ and $\varepsilon_{LIS}$ are 0.9 and 1.1, respectively. In our "perfect model" assumption, all the above values are unbiased, and the "true" ECS is 3.6°C.

We perform ECS calculations using Equation (6) with 10,000 Monte-Carlo draws to sample the uncertainty in forcings and explore impacts from different assumptions of climate forcing/efficacy and the ocean dynamical feedback (Figure 6). If we neglect the ocean dynamical feedback and assume that both GHG and LIS forcings have a unit efficacy, as has been done in most previous studies (e.g., Rohling et al., 2012), we obtain a median ECS of 4.5°C, an overestimate of 25% that is statistically distinguishable from uncertainties associated with climate forcings. Using the "true" efficacy of the LGM GHG or LIS

produces a small change in ECS (<~0.3°C) that approximately cancels each other, as $\varepsilon_{GHG}$ is smaller and $\varepsilon_{LIS}$ is larger than unity. Accounting for the ocean dynamical feedback greatly improves the ECS calculation, yielding a median of 3.5°C, 0.1°C smaller than the true ECS. In sum, this exercise highlights the importance of the ocean dynamical feedback, which, if neglected, may cause an overestimation of the ECS value using reconstructions of LGM forcings/responses.

## 5 Conclusions

In this study, we have quantified the radiative forcing and efficacy of LGM GHGs and LISs in CESM and examined the contribution of the ocean dynamical feedback to surface temperature changes by comparing simulations in fully coupled and slab ocean configurations. ERFs of LGM GHG and LIS are estimated to be –2.8 and –3.2 W m$^{-2}$, respectively. The efficacy of LGM GHG and LIS forcings are estimated to be 0.9 and 1.1, respectively, indicating that lowering GHGs to LGM levels is 10% less efficient in changing global temperature than that of doubling atmospheric $CO_2$ under PI conditions, while the LGM

LIS is 10% more efficient. The smaller-than-unity efficacy of LGM GHG forcing is primarily attributed to a smaller shortwave cloud feedback at lower temperatures, which is consistent with previous studies showing a temperature-dependent cloud feedback over high latitudes (e.g., Zhu & Poulsen, 2020). The greater-than-unity efficacy of LGM LIS forcing is caused by relatively larger temperature changes over land and the Northern Hemisphere subtropical oceans, which are linked to the LIS-induced wind changes and feedbacks in ocean-atmosphere coupling and clouds. Our calculations of LIS forcing and efficacy

account for the radiative effects from ice-sheet albedo and the topographic and dynamic effects associated with the ice-sheet elevation, in contrast to previous estimation that only considered the former. In addition, our simulations suggest that the effective radiative forcings and surface temperature responses of LGM GHG and LIS forcings are additive on the global level, which supports the approach in which individual forcing agents are considered separately.

Our simulations demonstrate that the full extent of LGM cooling cannot be realized if only fast feedbacks are accounted for. Overall, the slow ocean dynamical feedback amplifies the LGM cooling by 28% (from 5.3 to 6.8°C). LGM-based ECS calculations that neglect this ocean dynamical effects produce an overestimation by approximately 25%. In our simulations, the ocean dynamical feedback is primarily attributed to dynamical changes in the Southern Ocean, where a dynamical interaction between ocean stratification, mixed-layer heat flux convergence, and sea-ice cover is found to amplify the LGM cooling. Additionally, dynamical processes in the tropical oceans and the Atlantic also impact the regional and global

temperatures. Overall, our results suggest an important role of climate models in the quantification of climate forcings and efficacy and the ocean dynamical feedback.

**Code and data availability**

CESM model code is available through the National Center for Atmospheric Research software development repository (https://svn-ccsm-models.cgd.ucar.edu/cesm1/release_tags/cesm1_2_2_1/). Simulation data relevant to this work will be

published in the Zenodo repository (https://doi.org/10.5281/zenodo.3948405). Additional simulation data can be requested by contacting J.Z. (jiangzhu@ucar.edu).

**Author contributions**

JZ designed the study, performed the model simulation and analysis, and wrote the first draft of the manuscript. CJP contributed to the interpretation of results and the final draft.

**Competing interests**

The authors declare no competing interests.

**Funding**

CJP acknowledge funding support from the Heising-Simons Foundation grant #2016-05 and #2016-12 and NSF grant 2002397.

**Acknowledgements**

The authors thank Jessica Tierney for helpful discussion leading to improvement of the manuscript. Computing resources (doi:10.5065/D6RX99HX) were provided by the Climate Simulation Laboratory at NCAR's Computational and Information Systems Laboratory, sponsored by the National Science Foundation and other agencies.





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





**Table 1:** List of CESM simulations conducted in this study, including experiment name, climate forcing configurations, run length (years), GMST and GMST changes (°C), effective radiative forcing (ERF; W m$^{-2}$), and efficacy (unitless). One standard deviation calculated using 540 annual data is listed. Note that kernels-corrected ERF uses 12-month climatology and no uncertainty is provided.

| Experiment | GHG | LIS | Length | GMST or $\Delta$GMST | ERF$_{fsst}$ | ERF$_\lambda$ | ERF$_{kernel}$ | $\varepsilon$ |
|---|---|---|---|---|---|---|---|---|
| FCM_PI | PI | PI | 900+ | 15.1 | -- | -- | -- | -- |
| FCM_LGM | 21ka | 21ka | 900+ | −6.8 | -- | -- | -- | -- |
| ATM_PI | PI | PI | 30 | 14.9±0.03 | -- | -- | -- | -- |
| ATM_2CO2 | 2×PI | PI | 30 | +0.3±0.05 | 3.7±0.3 | +3.9±0.3 | +4.0 | -- |
| ATM_GHG | 21ka | PI | 30 | −0.2±0.04 | −2.6±0.2 | −2.8±0.3 | −2.8 | -- |
| ATM_LIS | PI | 21ka | 30 | −1.3±0.03 | −1.9±0.2 | −3.2±0.2 | -- | -- |
| ATM_LGM | 21ka | 21ka | 30 | −1.5±0.05 | −4.4±0.3 | −6.1±0.3 | -- | -- |
| SOM_PI | PI | PI | 60 | 14.9±0.06 | -- | -- | -- | -- |
| SOM_2CO2 | 2×PI | PI | 60 | +3.6±0.06 | -- | -- | -- | 1.00 |
| SOM_GHG | 21ka | PI | 60 | −2.2±0.11 | -- | -- | -- | 0.9±0.1 |
| SOM_LIS | PI | 21ka | 60 | −3.2±0.09 | -- | -- | -- | 1.1±0.1 |
| SOM_LGM | 21ka | 21ka | 60 | −5.3±0.09 | -- | -- | -- | 0.9±0.1 |



**Table 2:** Climate feedback parameters (units: W m$^{-2}$ K$^{-1}$) in the SOM simulations. Radiative kernels-based analysis is performed for SOM_2CO2 and SOM_GHG but not SOM_LIS due to the drastically different boundary conditions. APRP-based analysis is performed for
all three simulations. APRP quantifies the shortwave climate feedback parameter and decomposes the cloud feedback into contributions from changes in cloud amount, scattering, and absorption. The cloud absorption feedback is not shown, as it is small and varies litter between simulations. Values in parentheses are the contribution to the global mean value from ocean grid points in LGM simulation. Note that the high value in the Albedo column for SOM_LIS (1.31) includes the contribution from the shortwave forcing over land.

| Experiment | Radiative kernels | | | | | SW APRP | | | | |
|---|---|---|---|---|---|---|---|---|---|---|
| | Planck | Albedo | WV+LR | CLD$_{LW}$ | CLD$_{SW}$ | CLD | CLD amount | CLD scattering | Clear sky | Albedo |
| SOM_2CO2 | −3.57 | 0.42 | 1.51 | 0.13 | 0.33 | 0.39 (0.21) | 0.39 (0.19) | 0.08 (0.06) | 0.30 (0.16) | 0.33 (0.15) |
| SOM_GHG | −3.52 | 0.41 | 1.52 | 0.13 | 0.19 | 0.15 (0.10) | 0.37 (0.22) | −0.16 (−0.07) | 0.31 (0.17) | 0.39 (0.18) |
| SOM_LIS | -- | -- | -- | -- | | 0.17 (0.30) | 0.25 (0.24) | −0.05 (0.08) | 0.19 (0.11) | 1.31 (0.13) |






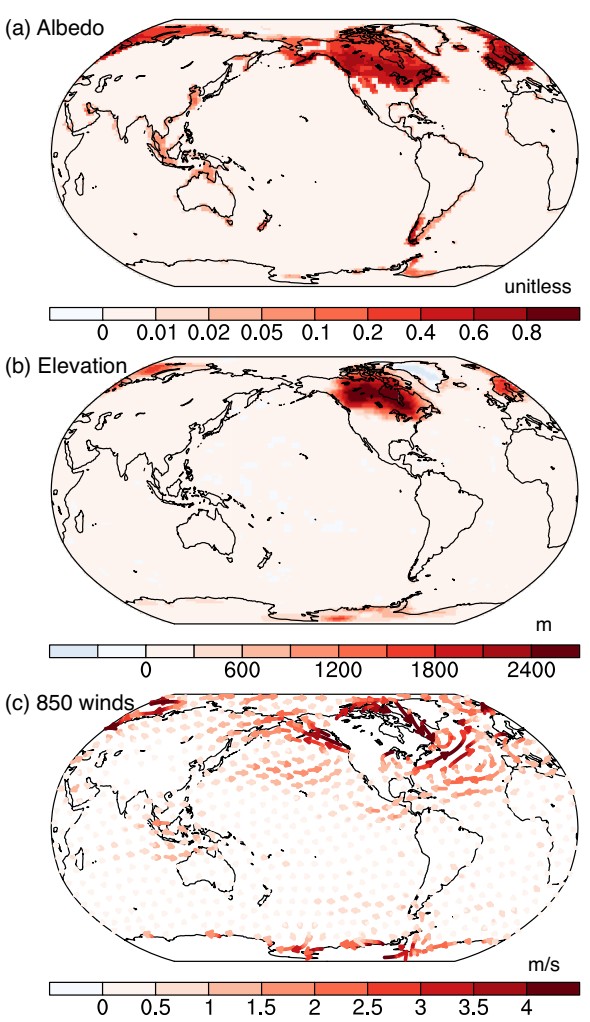

**Figure 1: (a)** Changes in shortwave surface albedo associated with the presence of the LGM ice sheets and shelf exposures due to the lowered sea level. Albedo changes are diagnosed using "fixed-SST" experiments (ATM_LIS and ATM_PI; see Section 2.2). Note the uneven colour bar. **(b)** Changes in surface elevation associated with the LGM ice sheets. **(c)** Wind changes at 850 hPa as an illustration of the ice-sheet dynamical forcing. Shown are anomalies in the "fixed-SST" experiments (ATM_LIS and ATM_PI).



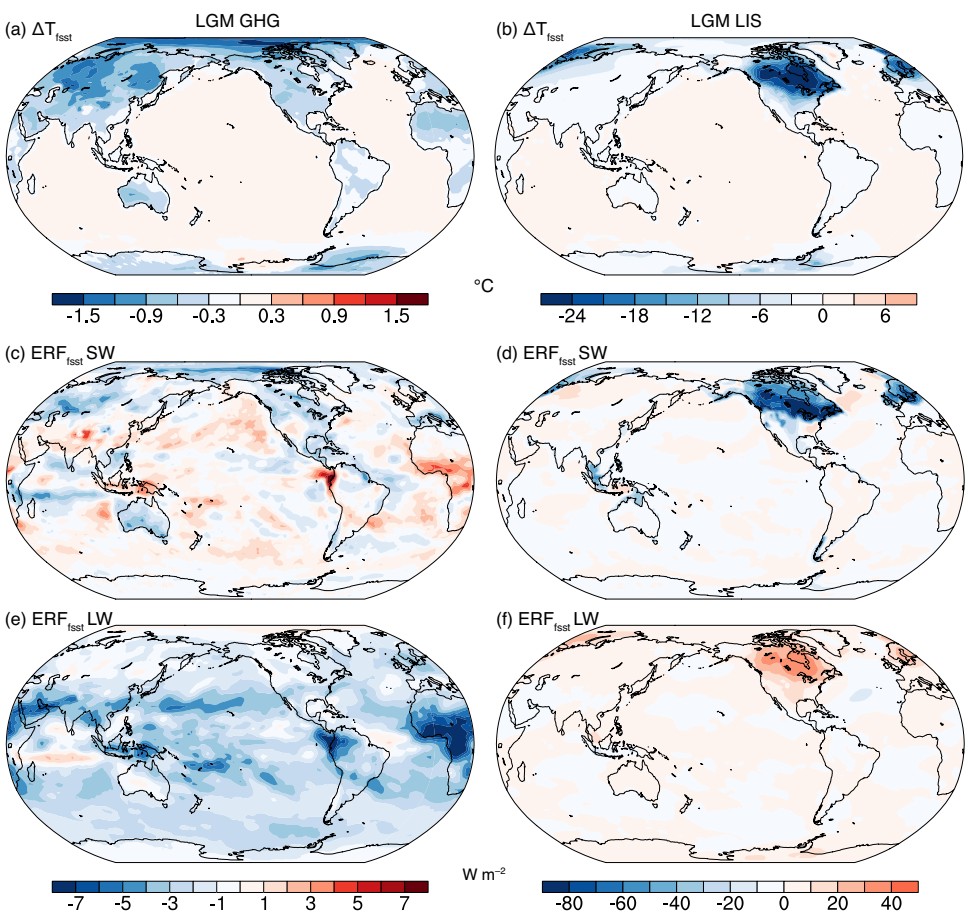

**Figure 2: (a)** Land surface temperature changes ($\Delta T_{fsst}$) in response to the LGM GHG forcing diagnosed in "fixed-SST" experiment ATM_GHG. **(b)** $\Delta T_{fsst}$ in response to the LGM LIS forcing diagnosed in ATM_LIS. **(c)** The shortwave component of effective radiative forcing associated with LGM GHG forcing diagnosed in ATM_GHG. **(e)** as **(c)**, but for the longwave component. **(d)** and **(f)**, as **(c)** and **(e)**, but for the LGM LIS forcing.




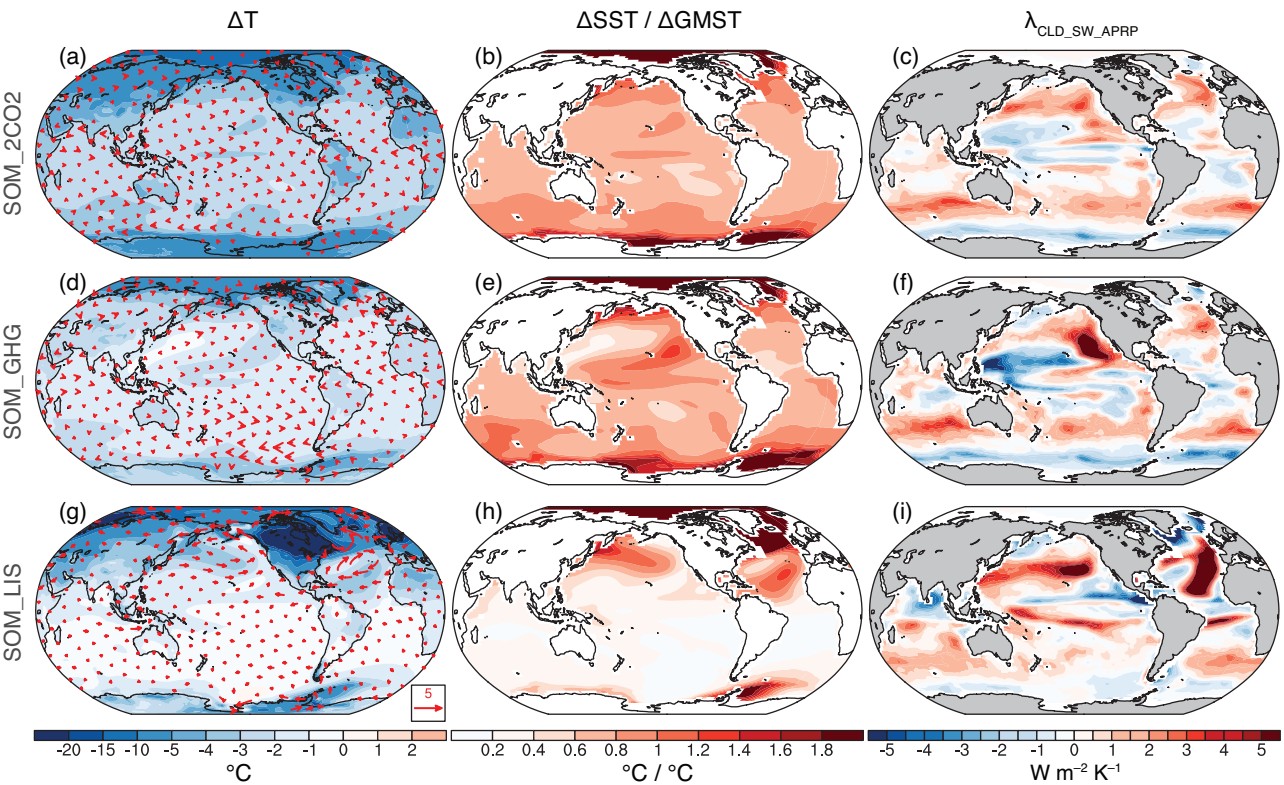

**Figure 3:** Changes in surface temperature in **(a)** SOM_2CO2, **(d)** SOM_GHG, and **(g)** SOM_LIS. Changes in 850-hPa winds (units: m s$^{-1}$) in fixed-SST experiments are shown as vectors in **(a)** ATM_2CO2, **(d)** ATM_GHG, and **(g)** ATM_LIS. Changes in SST scaled by the corresponding GMST change in **(b)** SOM_2CO2, **(e)** SOM_GHG, and **(h)** SOM_LIS. The shortwave cloud feedback parameter over LGM ocean grid points diagnosed using the APRP approach in **(c)** SOM_2CO2, **(f)** SOM_GHG, and **(i)** SOM_LIS. Note that temperature and wind changes in 2CO2 experiments in **(a)** have been multiplied by –1 to facilitate the comparison with those in SOM_GHG and SOM_LIS.


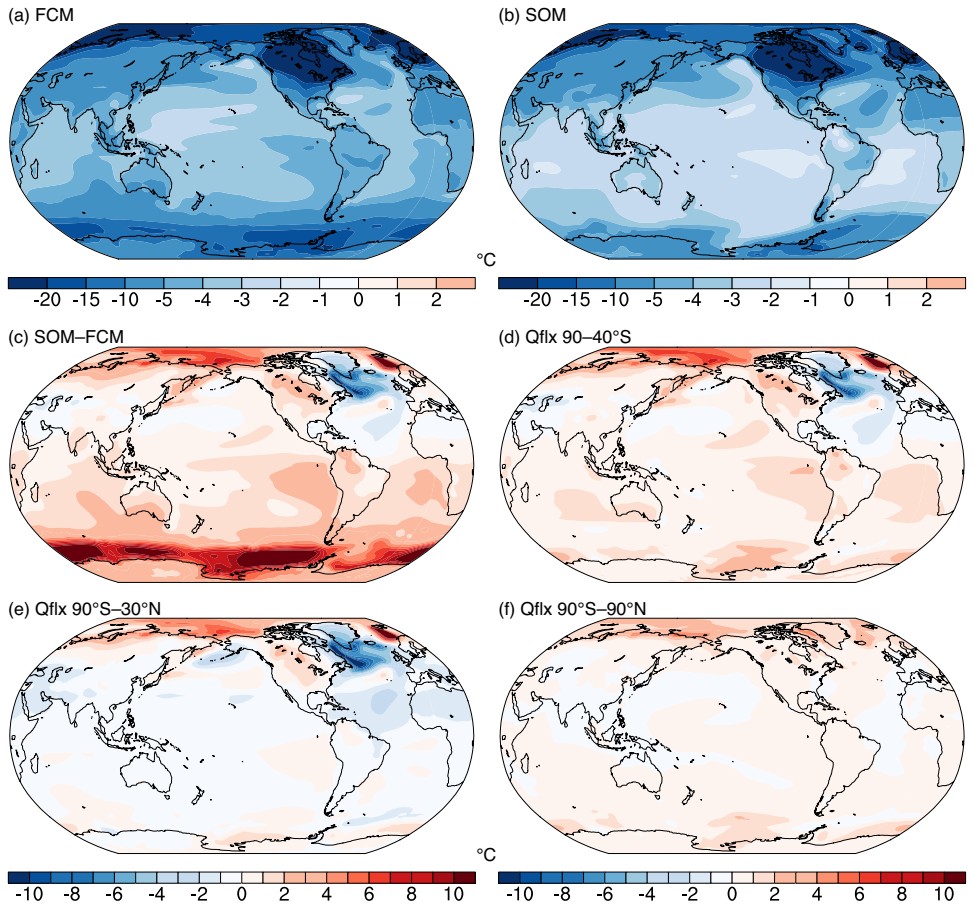

**Figure 4: (a)** LGM cooling in the fully coupled model simulations (FCM_LGM – FCM_PI). **(b)** LGM cooling in the slab ocean model simulations (SOM_LGM – SOM_PI). Both SOM_LGM and SOM_PI use the same ocean mixed layer depth and heat flux convergence (Qflx) that are derived from the fully coupled preindustrial simulation (FCM_PI). **(c)** Difference in the simulated LGM cooling between SOM and FCM (**(b)** – **(a)**). **(d)** as **(c)**, but for the SOM simulation with the prescribed Qflx over the Southern Ocean (90–40°S) replaced with that from FCM_LGM. **(e)** as **(c)**, but for the SOM simulation with the prescribed Qflx over 90°S–30°N replaced with that from FCM_LGM. **(f)** as **(c)**, but for the SOM simulation with the prescribed Qflx over the global ocean replaced with that from FCM_LGM. Note that a small intrinsic bias in surface temperature associated with SOM simulations (e.g. SOM_PI – FCM_PI) has been subtracted when comparing SOM and FCM simulations.





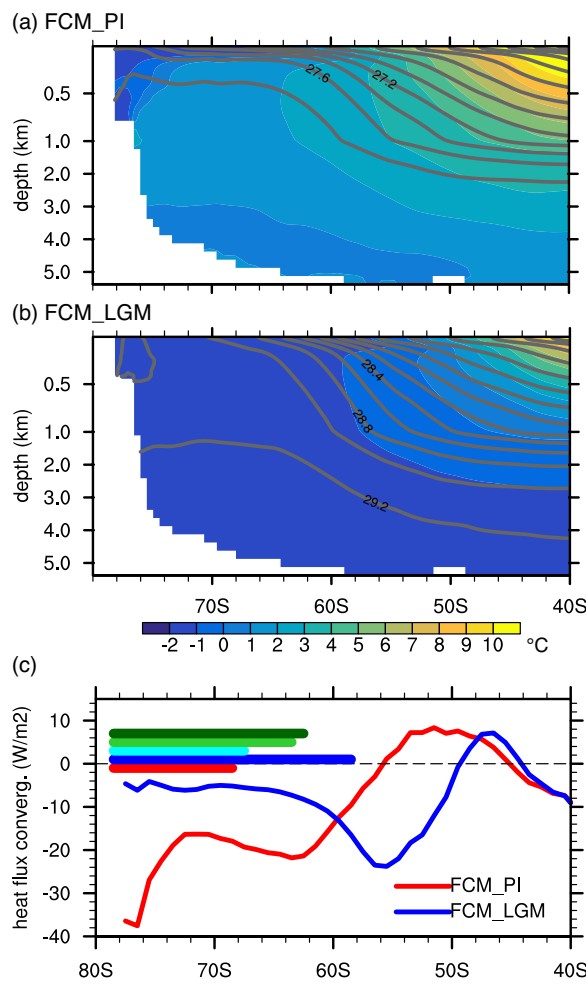

**Figure 5: (a)** Zonal mean potential temperature (shadings; units: °C) and potential density (contours) over the Southern Ocean in the fully
coupled preindustrial simulation (FCM_PI). Potential density values are reported in kg m$^{-3}$ subtracting 1000 kg m$^{-3}$. **(b)** as **(a)**, but for the
fully coupled LGM simulation (FCM_LGM). **(c)** Zonal mean heat flux convergence (units: W m$^{-2}$) in the ocean mixed layer in FCM_PI
(red) and FCM_LGM (blue). Negative values indicate ocean dynamical processes transport heat from below into the mixed layer. Horizonal
bars indicate the zonal mean semi-permanent sea-ice extent, defined as a 70% of the annual mean sea-ice cover, in the FCM_PI (red),
FCM_LGM (blue), SOM_LGM (cyan), and SOM simulations with Qflx over 90–40°S (light green) or 90°S–30°N (dark green) replaced
with that in FCM_LGM.




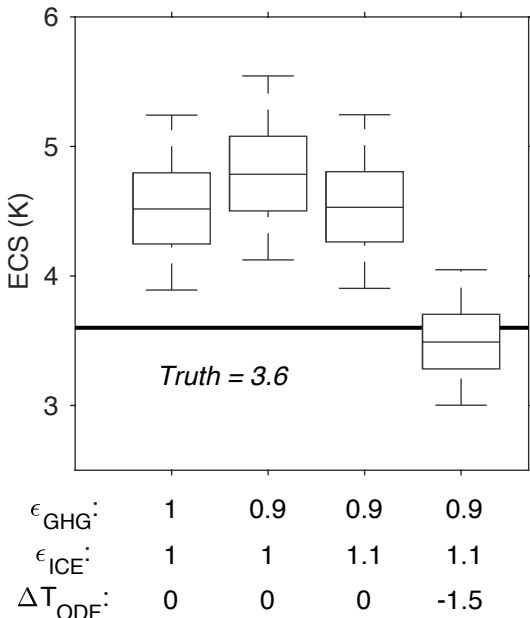

**Figure 6:** ECS in CESM1.2 (horizontal line) and the LGM-based estimates (box-and-whisker plots) using different values of forcing efficacy and ocean dynamical feedback (ODF)-induced ΔGMST changes. Each ECS estimation is obtained by performing 10,000 Monte-Carlo calculations, which incorporates the uncertainties (assumed to be Normal) in forcings and temperature responses. ERFs and efficacy of LGM GHG and LIS are listed in Table 1. ΔGMST changes from the ocean dynamical feedback is the difference between FCM and SOM simulations (Table 1).