# Peer review of "LGM climate forcing and ocean dynamical feedback and their implications for estimating climate sensitivity"

_Climate of the Past, 2020_

## Referee Comment (RC1) · Anonymous Referee #1 · 12 Aug 2020

This paper describes experiments performed with the CESM model and their analysis with respect to a deeper understanding of LGM climate and related calculations of equilibrium climate sensitivity (ECS). To my knowledge no such in-depth analysis has been performed so far, so I believe this is a great step forward.

I fully support publication and have mainly minor comments, dealing with details and representations. The discussion of the Southern Ocean stratification and of the resulting climate sensitivity need to get enhanced. However, my comments should be all addressable by the authors with little efforts.

Comments, in order of the written text:

1. lines 29–30: Please discuss GCM responses in the context of ECS analysis for longer than 100 years, e.g. based on LongRun-MIP (Rugenstein et al., 2019, 2020). Update the 1.5–4.5K range from the IPCC with results from the most recent review showing 2.6–3.9 K (66% confidence interval) (Sherwood et al., 2020).

2. line 38 and later: The reference "Rohling et al (2012)" should be cited as "PALAEOSENS-Project Members (2012)".

3. throughout the draft: The way papers with more authors are cited differs. Eg. line 43 says "Stap, Köhler & Lohmann, 2019", but the same paper is addressed as "Stap et al., 2019" later in line 59. Please homogenize the reference style to what is requested by the publisher.

4. line 52: To my knowledge "surface topography" should be included in Friedrich et al. (2016), please check.

5. line 53: What about sea level based elevation change? Figure 1b does not show that a sea level drop was considered (e.g. nothing seen in Sunda Shelf or Bering Strait). Maybe change color bar resolution for negative values.

6. line 74: The effect of sea level is not menioned here. Clarify if it is included or not.

7. lines 82–84: The list of past climate studies using this model is rather long, please reduce to some essentials or examples.

8. line 99: "a different atmosphere model". Which model?

9. line 102: It is stated that –6.8 K obtained in FCM_LGM is in agreement with Tierney et al. (2019). This is not true, since Tierney finds a LGM cooling of –5.9K (–6.3 to –5.6K, 95% CI), so what is found here is 0.5K below the 95% CI of Tierney.

10. line 123: Why are simulations run only for 30 years, if fast feedbacks should consider the changes of the first century? Can you state the final radiative forcing imbalance at TOA as a convergence criteria?

11. line 128ff: The climate sensitivity parameter in units $K\,W^{-1}\,m^2$ is given throughout the text as $\lambda$. I find this rather disturbing, because in most literature I am aware of $\lambda$ is used for the climate feedback parameter in units $W\,m^{-2}\,K^{-1}$, which is the inverse of the climate feedback parameter, e.g. to name a few PALAEOSENS-Project Members (2012); Köhler et al. (2010); Sherwood et al. (2020). I am aware that there is no agreed-upon way how to define variables here, but I believe the majority of the readers would be pleased if not $\lambda$ is taken as climate sensitivity parameter.

12. line 152: "60 years simulation time" again needs some more motivation, why not 100 years? If you make averages over the last 20 years of these 60 years, does this imply the TOA energy imbalance is already below 0.1 $W/m^2$ after 40 years, thus having only well equilibrated results to be averaged?

13. line 185: $\Delta T_{fSST}$ is 0.2deg C here, but I understand that SSTs are fixed and this averages over land only. If so, it should be negative, not positive, please check.

14. line 189: Labelling a scenario LGM_2CO2 is ok, but if you used 2CO2 in the text, you need to revise the writing to "$2\times CO_2$" or so.

15. line 191: should be "–24"deg C.

16. line 194: Expand if sea level fall is also part of the considered processes.

17. line 205: Does this ($ERF_\lambda$ from LGM LIS) include snow outside LIS regions and cloud adjustments (mainly due to sea level)?

18. line 215: "–3.3" misses a unit of "$W/m^2$"

19. lines 287ff: SO stratification: To my knowledge paleo data suggest a higher strati­fication in the Southern Ocean at LGM, while here a smaller one is found. See for example the vertical distribution of 13C in Figure 11 of Hodell et al. (2003). Maybe this has to do with difference in shallow and deep water stratification. There are also other papers discussing the role of Southern Ocean diapycnals or mixing and the variability of diapycnal mixing, e.g. see Jones and Abernathey (2019) or Hines et al. (2019) or Holzer et al. (2017) or Abernathey and Ferreira (2015). Please discuss how those processes are implemented in your model here with respect to what these papers suggest.

20. line 340: Using mean values and Eq 6 I obtain 3.4 K, not 3.6 K as stated in line 344. Maybe this is based on the effect of Monte-Carlo sampling which considers the uncertainties, maybe it is a typo, please check.

21. Discussion:

   (a) Please calculate out of your Monte-Carlo results and figure 6 and the range typcially given by others, e.g of 68% for $\pm 1\sigma$, or 95% $\pm 2\sigma$.

   (b) You find an ECS for LGM based on GHG, and land ice including efficacy of 3.6 K, so comparable to $S_{[GHG,LI]}$ in the nomenclature of PALAEOSENS-Project Members (2012), although this is the value BEFORE multiplying with the radiative forcing caused by "$2\times CO_2$". (Might also be called $S^{\varepsilon}_{[GHG,LI]}$ fol­lowing Stap et al. (2019) to account for the efficacy.) Please say so and also emphasis here (and in the conclusions) again, that vegetation and aerosols are kept fixed at PI level. Also note (and maybe compare), that $S_{[GHG,LI]}$ in PALAEOSENS-Project Members (2012) for LGM was $0.85 \pm 0.19$ K W$^{-1}$ m$^2$ which translates into an approximation of ECS of $3.3 \pm 0.4$K us­ing ERF_2CO2 of $3.9 \pm 0.3$ W m$^{-2}$ found here, so both studies are within uncertainities pretty much in agreement.

(c) line 346: "If we neglect the ocean dynamical feedback and assume that both GHG and LIS forcings have a unit efficacy, as has been done in most previous studies (e.g., Rohling et al., 2012)". This comparison to Rohling et al., 2012 is only correct for efficacy (no efficacy considered in Rohling). It is however incorrect for ocean dynamical feedback, since in Rohling temperature change is taken from data, which always include all processes, also vegetation and aerosols that has been kept fixed here.

(d) Stap et al. (2019) finds an efficacy of land ice of 0.45 with a large uncertainty. This is opposite and a lot more than the 1.1 found here. This need further discussions and suggestions for the differences.

(e) Furthermore, for $2\times$ $CO_2$ your model finds an ECS of $3.9 \pm 0.3$ K. The difference between this value and $S_{[GHG,LI]}$ @ LGM of 3.6 K indicate only a small state-dependency of ECS. If you include the uncertainties around 3.6 K obtained from the Monte-Carlo approach, they are more or less in agreement. This finding needs to get emphasised somewhere, as it seems to disagree with other findings, e.g. Crucifix (2006), for GCMs, but there are certainly newer papers, see also von der Heydt et al. (2014, 2016); Köhler et al. (2017). State-dependency is also discussed in the most recent review of Sherwood et al. (2020). Discuss you small state-dependency of ECS widely. What might be the reasons? The fixed vegetation and aerosols? The too cold state in the full climate model compared to data?

22. Figure 2a,b: I do not understand how temperature over ocean can be different from zero here, since I thought SST and sea ice have been fixed. There are blue shadings (negative values) in Arctic and Southern Ocean. Either explain or correct.

**References**

Abernathey, R. and Ferreira, D.: Southern Ocean isopycnal mixing and ventilation changes driven by winds, Geophysical Research Letters, 42, 10,357–10,365, doi:10.1002/2015GL066238, 2015.

Crucifix, M.: Does the Last Glacial Maximum constrain climate sensitivity?, Geophysical Research Letters, 33, L18 701, doi: 10.1029/2006GL027 137, 2006.

Friedrich, T., Timmermann, A., Tigchelaar, M., Elison Timm, O., and Ganopolski, A.: Nonlinear climate sensitivity and its implications for future greenhouse warming, Science Advances, 2, doi:10.1126/sciadv.1501923, 2016.

Hines, S. K., Thompson, A. F., and Adkins, J. F.: The Role of the Southern Ocean in Abrupt Transitions and Hysteresis in Glacial Ocean Circulation, Paleoceanography and Paleoclimatology, 34, 490–510, doi:10.1029/2018PA003415, 2019.

Hodell, D. A., Venz, K. A., Charles, C. D., and Ninnemann, U. S.: Pleistocene vertical carbon isotope and carbonate gradients in the South Atlantic sector of the Southern Ocean, Geochemistry, Geophysics, Geosystems, 4, 1004, doi: 10.1029/2002GC000 367, 2003.

Holzer, M., DeVries, T., Bianchi, D., Newton, R., Schlosser, P., and Winckler, G.: Objective estimates of mantle 3He in the ocean and implications for constraining the deep ocean circulation, Earth and Planetary Science Letters, 458, 305 – 314, doi:http://dx.doi.org/10.1016/j.epsl.2016.10.054, 2017.

Jones, C. S. and Abernathey, R. P.: Isopycnal Mixing Controls Deep Ocean Ventilation, Geophysical Research Letters, 46, 13 144–13 151, doi:10.1029/2019GL085208, 2019.

Köhler, P., Bintanja, R., Fischer, H., Joos, F., Knutti, R., Lohmann, G., and Masson-Delmotte, V.: What caused Earth's temperature variations during the last 800,000 years? Data-based evidences on radiative forcing and constraints on climate sensitivity, Quaternary Science Reviews, 29, 129–145, doi:10.1016/j.quascirev.2009.09.026, 2010.

Köhler, P., Stap, L. S., von der Heydt, A. S., de Boer, B., van de Wal, R. S. W., and Bloch-Johnson, J.: A state-dependent quantification of climate sensitivity based on paleo data of the last 2.1 million years, Paleoceanography, 32, 1102–1114, doi:10.1002/2017PA003190, 2017.

PALAEOSENS-Project Members: Making sense of palaeoclimate sensitivity, Nature, 491, 683–691, doi:10.1038/nature11574, 2012.

Rugenstein, M., Bloch-Johnson, J., Abe-Ouchi, A., Andrews, T., Beyerle, U., Cao, L., Chadha,

T., Danabasoglu, G., Dufresne, J.-L., Duan, L., Foujols, M.-A., Frölicher, T., Geoffroy, O., Gregory, J., Knutti, R., Li, C., Marzocchi, A., Mauritsen, T., Menary, M., Moyer, E., Nazarenko, L., Paynter, D., Saint-Martin, D., Schmidt, G. A., Yamamoto, A., and Yang, S.: LongRunMIP: Motivation and Design for a Large Collection of Millennial-Length AOGCM Simulations, Bulletin of the American Meteorological Society, 100, 2551–2570, doi:10.1175/BAMS-D-19-0068.1, 2019.

Rugenstein, M., Bloch-Johnson, J., Gregory, J., Andrews, T., Mauritsen, T., Li, C., Frölicher, T. L., Paynter, D., Danabasoglu, G., Yang, S., Dufresne, J.-L., Cao, L., Schmidt, G. A., Abe-Ouchi, A., Geoffroy, O., and Knutti, R.: Equilibrium Climate Sensitivity Estimated by Equilibrating Climate Models, Geophysical Research Letters, 47, e2019GL083 898, doi:10.1029/2019GL083898, 2020.

Sherwood, S., Webb, M. J., Annan, J. D., Armour, K. C., Forster, P. M., Hargreaves, J. C., Hegerl, G., Klein, S. A., Marvel, K. D., Rohling, E. J., Watanabe, M., Andrews, T., Braconnot, P., Bretherton, C. S., Foster, G. L., Hausfather, Z., Heydt, A. S. v. d., Knutti, R., Mauritsen, T., Norris, J. R., Proistosescu, C., Rugenstein, M., Schmidt, G. A., Tokarska, K. B., and Zelinka, M. D.: An assessment of Earth's climate sensitivity using multiple lines of evidence, Reviews of Geophysics, n/a, e2019RG000 678, doi:10.1029/2019RG000678, 2020.

Stap, L. B., Köhler, P., and Lohmann, G.: Including the efficacy of land ice changes in deriving climate sensitivity from paleodata, Earth System Dynamics, 10, 333–345, doi:10.5194/esd-10-333-2019, 2019.

von der Heydt, A. S., Köhler, P., van de Wal, R. S., and Dijkstra, H. A.: On the state dependency of fast feedback processes in (paleo) climate sensitivity, Geophysical Research Letters, 41, 6484–6492, doi:10.1002/2014GL061121, 2014.

von der Heydt, A. S., Dijkstra, H. A., van de Wal, R. S. W., Caballero, R., Crucifix, M., Foster, G. L., Huber, M., Köhler, P., Rohling, E., Valdes, P. J., Ashwin, P., Bathiany, S., Berends, T., van Bree, L., Ditlevsen, P., Ghil, M., Haywood, A., Katzav, J., Lohmann, G., Lohmann, J., Lucarini, V., Marzocchi, A., Pälike, H., Baroni, I. R., Simon, D., Sluijs, A., Stap, L. B., Tantet, A., Viebahn, J., and Ziegler, M.: Lessons on climate sensitivity from past climate changes, Current Climate Change Reports, 2, 148–158, doi:10.1007/s40641-016-0049-3, 2016.

---

## Referee Comment (RC2) · Anonymous Referee #1 · 13 Aug 2020

One point which I forgot in my review so far, but which might also get improved by more details:

The efficacy of GHG @ LGM was 0.9 here. I am wondering if the full impact of $CH_4$ on radiative forcing includes indirect effects of $CH_4$ on stratospheric $H_2O$ and tropospheric $O_3$. They have been proposed in Hansen et al., (2005) and they led to an additional change of 40% in radiative forcing of $CH_4$ (Hansen et al., 2008).

Hansen, J., Sato, M., Ruedy, R., Nazarenko, L., Lacis, A., Schmidt, G.A., Russell, G., Aleinov, I., Bauer, M., Bauer, S., Bell, N., Cairns, B., Canuto, V., Chandler, M., Cheng, Y., Genio, A.D., Faluvegi, G., Fleming, E., Friend, A., Hall, T., Jackman, C., Kelley,

M., Kiang, N., Koch, D., Lean, J., Lerner, J., Lo, K., Menon, S., Miller, R., Minnis, P., Novakov, T., Oinas, V., Perlwitz, J., Perlwitz, J., Rind, D., Romanou, A., Shindell, D., Stone, P., Sun, S., Tausnev, N., Thresher, D., Wielicki, B., Wong, T., Yao, M., Zhang, S., 2005. Efficacy of climate forcings. Journal of Geophysical Research 110, D18104. doi:10.1029/2005JD005776.

Hansen, J., Sato, M., Kharecha, P., Beerling, D., Berner, R., Masson-Delmotte, V., Pagani, M., Raymo, M., Royer, D.L., Zachos, J.C., 2008. Target atmospheric CO2: where should humanity aim? The Open Atmospheric Science Journal 2, 217–231. doi:10.2174/1874282300802010217.

---

## Referee Comment (RC3) · Andreas Schmittner (Referee) · 22 Aug 2020

The authors present climate model simulations and analysis regarding climate sensitivity, radiative forcing and feedbacks in the LGM. Simulations with fixed SSTs and a slab ocean model and a model with a fully dynamic ocean component are used to analyze forcing from greenhouse gases and ice sheets. Differences between the fixed SST and slab ocean model results give "effective radiative forcings" and "efficacy" of forcings. The method is quite complicated (at least for a person not intimately familiar with these concepts). This also makes for difficult reading. I find it a little confusing that some of the so called forcings include what would be traditionally considered feedbacks, e.g. the longwave response to temperature changes illustrated in Fig. 2 would have traditionally been considered a feedback. But I guess this is the purpose, to separate effects beyond the traditional forcing/feedbacks concept. Despite the difficulty for a general reader I think technical papers like this are important because they advance the science by adding more detailed information. For this reason, I support the publication of this paper, perhaps in a slightly modified form.

An important conclusion is the ocean dynamical feedback, which the authors suggest increases climate sensitivity.

This is a pure model analysis without consideration of observations. This is fine, but I think the paper could benefit from some additional discussion regarding observations. E.g. the shortwave forcing from ice sheets certainly depends on the cloud cover simulated in the preindustrial (PI) model over regions that become ice covered in the LGM. E.g. if cloud cover and thus planetary albedo is high already at PI it won't change as much by adding ice compared with a low cloud cover/albedo case. I think this is a major source of uncertainty in determining the LGM shortwave forcing from ice sheets. It could be addressed by comparing model's cloud cover/albedo with present day observations in those regions. So, I suggest the authors to think about this and perhaps include a discussion about it in a revised version. If possible, a model data comparison would be useful, but it is not necessary. (My philosophy of reviewing papers is that the reviewer should not demand additional evidence, but evaluate the evidence presented.)

Another more major issue is the effect of sea level drop. It is not mentioned in the paper, but I think it should be, because the topographic effect of adding ice to the continents is accompanied by the effect of lower sea levels. Was this considered in the simulations presented here? In other words, was atmospheric mass conserved?

Another issue is the vegetation response, which apparently is fixed at present day. This will presumably affect land temperatures and thus the calculations presented in the paper. A discussion is warranted. E.g. it is well known that during the LGM much

of the present day boreal forest was converted to tundra. This would affect the surface albedo not only directly but also by modifying the effect of snow on the surface albedo.

Minor comments: numbers indicate line numbers 55: consider including Broccoli 2000 J. Clim. 13, 951pp and Schmittner et al., 2011 doi: 10.1126/science.1203513

102-103: The authors compare their model with new results from Tierney et al., which is fine, but I think they should also compare to previous results e.g. the MARGO SST compilation. Also, I think there is some circularity here because I think Tierney et al. used the CESM model data for their temperature estimates, so they are not independent. I'd also suggest comparing their model's whole ocean temperature change with data from Bereiter et al., (2018, Nature 553, 39–44). It may also be useful to show or note the trend in whole ocean T if not in equilibrium.

105: Sherwood et al. should be (2015) I think

111: "active land model" not quite correct if vegetation is fixed

127: is the correction applied locally or globally?

225: "over high-latitude regions" Looking at Fig. 3c,f, I don't see large changes at high latitudes. I see more negative values for lambda over the western tropical Pacific and more positive values over the eastern subtropical Pacific. Would be useful to elaborate on this more.

234: "0.31" shouldn't this be 0.1? Does it refer to the CLD column in SOM\_GHG?

252: typo: the "importance"

296: consider adding Schmittner 2003 EPSL doi:10.1016/S0012-821X(03)00291-7

311: add "in the model" to clarify that this statement refers to the model, not the real world

362: "high latitudes" see my comment above

---

## Editor Comment (EC1) · Luke Skinner (Editor) · 22 Sep 2020

Dear Jiang Zhu,

You will have noticed that two sets of review comments have now been received for your recent submission to Climate of the Past. The comments are generally conducive to the continuation of the review process, and I therefore invite you to provide a suitably revised manuscript for consideration, along with a detailed point-by-point response to the reviewer comments. I would also like to take this opportunity to provide a few comments of my own, primarily of an editorial nature, which you may wish to also consider in preparing your revised manuscript.

Editorial comments:

1. The concept of equilibrium climate sensitivity (ECS) is defined very clearly the introduction; however, I wonder if (for the benefit of some readers who may be more exclusively familiar with the palaeoclimate context) it might be useful to disambiguate 'ECS' from the term 'ES', designating 'Earth System Sensitivity', which has been introduced in the literature (e.g. by Schneider et al., Lund et al.) and which aims to include the effects of slower feedbacks in the climate system. This is just a suggestion of course; the definition that is provided for ECS per se is very clear.

2. Line 46: reference is made to the term 'efficacy' here, and elsewhere in the introductory text; however it is only clearly defined on line 162 and in equation 4. I would suggest that a brief verbal definition of the term be provided up front (e.g. along the lines of "the ratio of the climate sensitivity parameter (K/wm-2) derived for a given forcing relative to that for a doubling of atmospheric CO2", or "the ratio of the warming effect attributed to a given forcing, relative to that due to a doubling of atmospheric CO2 under pre-industrial conditions".. etc. . .). I realise that it might be hard to be succinct and accurate at the same time, but this is an important concept in the paper and it will be important to make it clear to readers early on.

3. Line 73: I found this sentence hard to decipher, and wondered if the following was an accurate reflection of what was intended: ". . .to provide a complete quantification of the LGM LIS and GHG forcing, and their respective 'efficacies', using a suite of climate simulations."

4. Line 117: It is not clear what the last sentence means to say; please rephrase to clarify (e.g. do adjustments reflect changes that occur 'as a direct result of a given forcing, without mediation by global average temperature change, i.e. not including the Planck feedback?'). It is hard to see immediately what changes in temperature, clouds etc. . ., would be mediated by 'global average' temperature change specifically, as opposed to local/regional changes, apart from the Planck feedback on global longwave

output. For example, I understand that sea-ice and snow cover changes that arise from a cooling caused by a GHG change would be excluded as 'adjustments', but do these really arise from 'global average' temperature changes?

5. Line 160: Is it possible to clarify this sentence? E.g. "...represents the global surface air T change associated with an effective radiative forcing, but that is driven indirectly (by SST change)"? Is my suggestion accurate?

6. Linen 214: As I will expand upon a little more below, I find the phrasing 'overestimation/underestimation' somewhat misleading at times, or at least open to misunderstanding. For example, here, I would suggest that it might be clearer to state something like: "...this APRP approach overestimates the shortwave radiative forcing that is attributable exclusively to changes in LIS extent, as it includes the radiative effect of snow increases over ice sheets (or regions with shelf exposure); the albedo of fresh snow is considerably larger...."

7. Line 216: Similarly I would suggest a minor clarification such as: "The snow-induced overestimation [of the LIS direct contribution] is larger if the cooling over ice sheets is greater."

8. Line 218: I think that the use of plural for simulations might be better, i.e.: "...is greater in coupled simulations... atmosphere-only simulations...

9. Line 224: Is the study of M. Crucifix (2006, Does the Last Glacial Maximum constrain climate sensitivity?, Geophys. Res. Lett., 33, L18701) relevant here at all (with respect to the temperature dependence of cloud feedbacks)?

10. Line 258: "...the importance of using...". I would also suggest adding for clarity: "...using efficacy to evaluate the overall effectiveness of their radiative forcing as compared to a doubling of atmospheric CO2."

11. Section 3.3: The point here seems to be that the system is broadly linear (at least by virtue of any regional non-linearities cancelling out globaly perhaps?); however, I

wondered if it would it be justified in your view to add a caveat that this point applies primarily to an evaluation of short-term impacts (i.e. from fast feedbacks)?

12. Line 273: Surely ocean interior temperatures will not be in equilibrium after 60 years, or if they are in the SOM some caution is warranted in extrapolating to the real global ocean? I simply invite your consideration of whether any clarification is needed here.

13. Line 289: Would it be more complete to state that the remote impact on the SO reflects the impact on SO stratification of a displacement in tropical atmospheric circulations, etc..?

14. Line 333: Again, can I suggest to add the clarification: "We note that, due to the inclusion of snow effects in the forcing quantification, the APRP-based approach overestimates the direct shortwave albedo effects that are attributable only to changing LIS extent"? My point is that it is only an 'overestimation' if one wants to strip out the knock-on effects of a changing LIS, to consider only direct impacts. Otherwise, one could argue, conversely that the 'real' impact of changing LIS extent is actually underestimated by an approach that does not consider the knock-on effects.

15. Line 352: Here again I would suggest to alter slightly the language used, for clarity. E.g. "If we do not remove the ocean dynamical feedbacks...".

16. Line 357: Similarly, can I suggest for your consideration: "In sum, this exercise highlights the importance of the ocean dynamical feedback, which, if included, may cause an overestimation of the ('fast feedback') ECS value using reconstructions of LGM forcings/responses." To my mind, 'neglecting' the ocean dynamical feedback would be the same as not stripping it out of the radiative/temperature effects, which is somewhat confusing.

17. Line 375: In the same vein as the above comments, can I propose for you to consider: "LGM-based ECS calculations that neglect to remove ocean dynamical effects

produce an overestimation [of fast feedbacks/sub-centennial impacts] by approximately 25%." My point is simply that I it may be important to make sure no one misunderstand this statement as suggesting that the ocean dynamical feedback dampens warming, when in fact it amplifies it.

18. Finally, I can't help but add to Referee 1's comment number 19, that radiocarbon evidence from the LGM is likely more useful as a constraint on large-scale mixing/air-sea exchange of heat/carbon than is d13C, which notably has a non-conservative component due to biological export production. In any event, both lines of evidence would indeed suggest greater stratification/sea ice coverage, not less.

I greatly look forward to receiving your revised manuscript and response to the reviewer comments.

Sincerely, Luke Skinner

---

## Author Comment (AC1) · 1 Dec 2020

This paper describes experiments performed with the CESM model and their analysis with respect to a deeper understanding of LGM climate and related calculations of equilibrium climate sensitivity (ECS). To my knowledge no such in-depth analysis has been performed so far, so I believe this is a great step forward.

I fully support publication and have mainly minor comments, dealing with details and representations. The discussion of the Southern Ocean stratification and of the resulting climate sensitivity need to get enhanced. However, my comments should be all addressable by the authors with little efforts.

**Reply:** We thank Reviewer 1 for the positive assessment of our manuscript. In the revised manuscript, we will also address all the insightful comments.

Comments, in order of the written text:

1. lines 29–30: Please discuss GCM responses in the context of ECS analysis for longer than 100 years, e.g. based on LongRun-MIP (Rugenstein et al., 2019, 2020). Update the 1.5–4.5K range from the IPCC with results from the most recent review showing 2.6–3.9 K (66% confidence interval) (Sherwood et al., 2020).

**Reply:** Thanks for the suggestion. We note that we do not discuss models and model-based ECS estimates in the first paragraph of Introduction. Therefore, we will not mention LongRun-MIP results in Introduction.

We will revise the sentence as follows: "ECS range is estimated to be 2.6–3.9°C (66% confidence interval) in a recent assessment (Sherwood et al., 2020), which represents a narrower range than the traditional one of 1.5–4.5°C (IPCC, 2013)."

2. line 38 and later: The reference "Rohling et al (2012)" should be cited as "PALAEOSENS-Project Members (2012)".

**Reply:** We will make this change in the revised manuscript.

3. throughout the draft: The way papers with more authors are cited differs. Eg. line 43 says "Stap, Köhler & Lohmann, 2019", but the same paper is addressed as "Stap et al., 2019" later in line 59. Please homogenize the reference style to what is requested by the publisher.

**Reply:** The citation is automatically generated following the APA style. We believe this will be homogenized during typeset by the publisher.

4. line 52: To my knowledge "surface topography" should be included in Friedrich et al. (2016), please check.

**Reply:** We have checked and found that Friedrich et al. (2016) did not consider temperature change that is directly attributed to changes in surface elevation, i.e., "surface" is defined at different elevation between the preindustrial and LGM.

5. line 53: What about sea level-based elevation change? Figure 1b does not show that a sea level drop was considered (e.g. nothing seen in Sunda Shelf or Bering Strait). Maybe change color bar resolution for negative values.

**Reply:** Sea level-based elevation change is on the order of ~100 m and the effect has been included as changes in land-sea mask in our simulations. This can be seen in the surface albedo change (including Sunda Shelf and Bering Strait) in Figure 1a.

6. line 74: The effect of sea level is not mentioned here. Clarify if it is included or not.

**Reply:** Thanks. Our simulations have accounted for the effect from sea-level change between the preindustrial and LGM. We will clarify this in "Method, model, and experiments" by adding the following sentence: "LIS forcing is derived from the ICE-6G reconstruction (Peltier, Argus, & Drummond, 2015) and includes changes in land elevation and surface properties due to the presence

of LGM ice sheets, as well as changes in the land-sea mask to account for the lower sea level at the LGM".

7. lines 82–84: The list of past climate studies using this model is rather long, please reduce to some essentials or examples.

**Reply:** The list will be shortened.

8. line 99: "a different atmosphere model". Which model?

**Reply:** We meant the change of the atmosphere model from CAM4 to CAM5. We will clarify this in the revised manuscript by saying "an updated atmosphere model".

9. line 102: It is stated that –6.8 K obtained in FCM_LGM is in agreement with Tierney et al. (2019). This is not true, since Tierney finds an LGM cooling of – 5.9K (–6.3 to –5.6K, 95% CI), so what is found here is 0.5K below the 95% CI of Tierney.

**Reply:** The is a good point. Tierney provided two estimates: a data assimilation result of –6.1°C (–6.5 to –5.7) and a data-only result of –5.6°C (–6.8 to –4.4). We used the data-only result for the model-data comparison, as the data assimilation results contains information from CESM.

We will change the sentence as follows: "GMST in FCM_LGM is 6.8°C lower than that in FCM_PI and falls within the range directly estimated from proxy data in a recent study (–6.8 to –4.4°C; Tierney et al., 2020)."

10. line 123: Why are simulations run only for 30 years, if fast feedbacks should consider the changes of the first century? Can you state the final radiative forcing imbalance at TOA as a convergence criteria?

**Reply:** We run the "fixed-SST" atmosphere-only simulations for 30 years, following the Radiative Forcing Model Intercomparison Project protocol (e.g., see Pincus, Forster, & Stevens, Geosci. Model Dev., 9, 3447–3460, 2016). 30 years are long enough to remove the internal climate variability and produce robust estimates of effective radiative forcing, especially when SSTs and sea ice are prescribed. For the reasons that SSTs are fixed in the simulation, the simulations do not have a convergence problem and therefore the TOA imbalance is not shown.

11. line 128ff: The climate sensitivity parameter in units $K \, W^{-1} \, m^2$ is given throughout the text as $\lambda$. I find this rather disturbing, because in most literature I am aware of $\lambda$ is used for the climate feedback parameter in units $W \, m^{-2} \, K^{-1}$, which is the inverse of the climate feedback parameter, e.g. to name a few PALAEOSENS- Project Members (2012); Köhler et al. (2010); Sherwood et al. (2020). I am aware that there is no agreed-upon way how to define variables here, but I believe the majority of the readers would be pleased if not $\lambda$ is taken as climate sensitivity parameter.

**Reply:** We thank Reviewer 1 for the suggestion. We will change the notation and use $\alpha$ as the climate sensitivity parameter in the revised manuscript.

12. line 152: "60 years simulation time" again needs some more motivation, why not 100 years? If you make averages over the last 20 years of these 60 years, does this imply the TOA energy imbalance is already below 0.1 W/m2 after 40 years, thus having only well equilibrated results to be averaged?

**Reply:** Slab ocean simulations use prescribed effects from ocean dynamics and reaches equilibrium in less than 50 years; this timescale is determined by the thermal inertia of the ocean mixed layer (Bitz et al., 2011; Danabasoglu & Gent, J. Climate, 22(9), 2494–2499, 2009). In our analysis, we used last 20 years, which has an average TOA energy imbalance less than 0.1 $Wm^{-2}$. Results do not depend on whether last 20 or 10 years are used for analysis with a difference less than 5%.

13. line 185: $\Delta T_{fSST}$ is 0.2deg C here, but I understand that SSTs are fixed and this averages over land only. If so, it should be negative, not positive, please check.

**Reply:** We will make the correction in the revised manuscript.

14. line 189: Labelling a scenario LGM_2CO2 is ok, but if you used 2CO2 in the text, you need to

revise the writing to "2×CO2" or so.

**Reply:** Thank you for the suggestion. We will make the correction in the revised manuscript.

15. line 191: should be "–24"deg C.

**Reply:** This will be corrected.

16. line 194: Expand if sea level fall is also part of the considered processes.

**Reply:** The "increased coverage of land" in the manuscript refers to the shelf exposures due to the lowered sea level. We will add clarification in the revised manuscript.

17. line 205: Does this (ERF$\lambda$ from LGM LIS) include snow outside LIS regions and cloud adjustments (mainly due to sea level)?

**Reply:** Yes, $ERF_\lambda$ (Equations 1 and 3) here contains effects of atmosphere adjustments (including cloud changes). Accounting for these adjustments makes $ERF_\lambda$ more consistent with the framework of forcing and response (Sherwood et al., 2015). Effect of snow outside LIS on $ERF_{fsst}$ is relatively small outside LIS regions in the simulation with prescribed preindustrial SST; this can be partly seen from Figure 2d, which does not show large negative shortwave $ERF_{fsst}$ outside LIS regions. Importantly, this snow effect is largely removed when correcting the land surface temperature changes in "fixed-SST" simulations (Equation 1). This interpretation is supported by the fact that $ERF_\lambda$ nearly equals $ERF_{kernel}$ for both LGM GHG and $2\times CO_2$ (Table 1). It suggests that $ERF_\lambda$ effectively removes effects from changes in snow albedo, air temperature, and water vapor over land, as the kernel method does.

18. line 215: "–3.3" misses a unit of "W/m2"

**Reply:** Units will be added.

19. lines 287ff: SO stratification: To my knowledge paleo data suggest a higher stratification in the Southern Ocean at LGM, while here a smaller one is found. See for example the vertical distribution of 13C in Figure 11 of Hodell et al. (2003). Maybe this has to do with difference in shallow and deep water stratification. There are also other papers discussing the role of Southern Ocean diapycnals or mixing and the variability of diapycnal mixing, e.g. see Jones and Abernathey (2019) or Hines et al. (2019) or Holzer et al. (2017) or Abernathey and Ferreira (2015). Please discuss how those processes are implemented in your model here with respect to what these papers suggest.

**Reply:** Our LGM simulation exhibits decreased *shallow water* stratification over the deep-water formation region underneath sea ice in the Southern Ocean (*southward of ~60°S and above ~500m*; Figure 5a, b). The upper-ocean stratification is the most relevant for the mixed layer heat convergence and interactions with sea ice.

Major features of deep-ocean circulation and seawater characteristics in our CESM1 simulations agree well with findings from proxy reconstructions (e.g Hodell et al., 2003; Curry and Oppo, 2005; Adkins et al., 2002). These features include an expansion of the Antarctic Bottom Water, a shallower North Atlantic Deepwater, an increase in abyssal stratification, and a saltier and colder southern-source deep water. The role of a dynamical coupling between sea ice and the ocean stratification is also consistent with modeling and theoretical studies (e.g., Shin et al., 2003; Ferrari et al., 2014; Abernathey and Ferreira, 2015).

We will add a short discussion on the above aspects, as the focus of this study is not on examining details of the LGM ocean circulation. We will also clarify in the revised manuscript that we are referring to the upper-ocean stratification, which is more relevant to the heat budget within the ocean mixed layer and the expansion of sea ice.

20. line 340: Using mean values and Eq 6 I obtain 3.4 K, not 3.6 K as stated in line 344. Maybe this is based on the effect of Monte-Carlo sampling which considers the uncertainties, maybe it is a typo, please check.

**Reply:** There is some confusion here. Line 344 reads "In our "perfect model" assumption, all the above values are unbiased, and the "true" ECS is 3.6°C." The value of 3.6 K is the "truth" in the "perfect model" scenario in CESM1.2, obtained through performing 2×CO2 simulations. We think this is clearly written in the manuscript and, therefore, have left the value as it is.

21. Discussion:

(a) Please calculate out of your Monte-Carlo results and figure 6 and the range typically given by others, e.g of 68% for ±1σ, or 95% ±2σ.

**Reply:** We will adopt your suggestion and redo the Monte-Carlo calculation and Figure 6 in the revised manuscript.

(b) You find an ECS for LGM based on GHG, and land ice including efficacy of 3.6 K, so comparable to S[GHG,LI] in the nomenclature of PALAEOSENS-Project Members (2012), although this is the value BEFORE multiplying with the radiative forcing caused by "2×CO2". (Might also be called Sε following Stap et al. (2019) to account for the efficacy.) Please say so and also emphasis here (and in the conclusions) again, that vegetation and aerosols are kept fixed at PI level. Also note (and maybe compare), that S[GHG,LI] in PALAEOSENS-Project Members (2012) for LGM was 0.85 0.19 K W−1 m2 which translates into an approximation of ECS of 3.3±0.4K using ERF_2CO2 of 3.9±0.3 W m−2 found here, so both studies are within uncertainties pretty much in agreement.

**Reply:** We stress that the goal of this modeling study is not to estimate ECS but to examine the methodology that have been used to estimate ECS using paleoclimate reconstructions. In the model, we can directly obtain ECS through performing model simulations with the 2×CO2 forcing.

The apparent agreement of our ECS in calculation (Equation 6) and in CESM1.2 with results (~3.3°C) from PALAEOSENS-Project Members (2012) is caused by cancellation errors. PALAEOSENS-Project Members (2012) used a LIS forcing of –4.5 W m$^{-2}$ and an LGM cooling of –6.1 to –5.1°C, assumed unity efficacies of GHG and LIS forcings, and did not account for the ocean dynamical feedback. In contrast, our CESM1.2 simulation suggests a larger LGM cooling of –6.8°C, a smaller LIS forcing of –3.2 W m$^{-2}$, non-unity efficacy of both GHG and LIS, and a large contribution of ocean dynamics to the magnitude of LGM temperature change.

(c) line 346: "If we neglect the ocean dynamical feedback and assume that both GHG and LIS forcings have a unit efficacy, as has been done in most previous studies (e.g., Rohling et al., 2012)". This comparison to Rohling et al., 2012 is only correct for efficacy (no efficacy considered in Rohling). It is however incorrect for ocean dynamical feedback, since in Rohling temperature change is taken from data, which always include all processes, also vegetation and aerosols that has been kept fixed here.

**Reply:** Our simulations suggest that 28% of the LGM cooling is caused by ocean dynamics. This additional LGM cooling is achieved through a dynamical ocean-sea ice coupling, which produces a stronger sea-ice albedo feedback (see Figure 5c for the sea-ice cover in SOM and FCM simulations). In contrast, in response to 2×CO2, the sea-ice albedo feedback is much weaker than that in the LGM simulation (see Kutzbach et al., 2013). The results, therefore, suggest that the LGM cooling obtained from data is not fully caused by fast feedbacks that are operating in the 2×CO2 scenario. A correct way would be to subtract the ocean dynamics-induced temperature change before estimating ECS, or to account for the non-constant fast feedbacks that depend on ocean dynamics.

In the revised manuscript, we will clarify that the ocean dynamical effects on temperature is achieved through changing fast feedbacks.

(d) Stap et al. (2019) finds an efficacy of land ice of 0.45 with a large uncertainty. This is opposite and a lot more than the 1.1 found here. This need further discussions and suggestions for the differences.

**Reply:** We will add a discussion on the difference in LIS efficacy between our CESM and Stap et al. (2019) results, but we point out that a precise explanation is impossible considering the differences in the definition of forcing/efficacy, models, and experimental design. Specifically, Stap et al. (2019) used the compilation of model results of ω (the relative impact of land ice changes on the LGM

temperature anomaly) from Shakun (2017); most of these results are from intermediate complexity models. Our CESM results suggest an important role of the cloud feedback in determining the LGM temperature response (Figure 3i and Tables 1 and 2). It is unclear how these cloud processes are resolved in models with intermediate complexity.

We will add the following sentence in the revised manuscript: "Our simulations suggest an LGM LIS efficacy of 1.1, which differs from the 0.45 in Stap et al. (2019). A precise explanation about this difference is challenging, given the large differences in the definition of forcing/efficacy, model complexity, and experimental design."

(e) Furthermore, for 2CO2 your model finds an ECS of 3.9±0.3 K. The difference between this value and S[GHG,LI] @ LGM of 3.6 K indicate only a small state-dependency of ECS. If you include the uncertainties around 3.6 K obtained from the Monte-Carlo approach, they are more or less in agreement. This finding needs to get emphasised somewhere, as it seems to disagree with other findings, e.g. Crucifix (2006), for GCMs, but there are certainly newer papers, see also von der Heydt et al. (2014, 2016); Köhler et al. (2017). State-dependency is also discussed in the most recent review of Sherwood et al. (2020). Discuss you small state-dependency of ECS widely. What might be the reasons? The fixed vegetation and aerosols? The too cold state in the full climate model compared to data?

**Reply:** Our results suggest a strong state dependence of ECS, which manifests in the ocean dynamical effects and the non-unity efficacy (0.9) of LGM GHG forcing. The ocean dynamical effect depends on background climate and its effect on surface temperature is achieved through changing fast feedbacks (see also our response above). In the revised manuscript, we will clarify this point.

22. Figure 2a,b: I do not understand how temperature over ocean can be different from zero here, since I thought SST and sea ice have been fixed. There are blue shadings (negative values) in Arctic and Southern Ocean. Either explain or correct.

**Reply:** Thanks for pointing this out. The land-sea mask shown in the figures are from the present-day land-sea distribution, so some of the negative values are caused by the fast that they are ocean grid points at present day but land grid points at the LGM. Also, surface temperature above sea ice is free to change, as we only fix the SST and sea-ice cover in the simulations. We will clarify this in the figure caption in the revised manuscript.

**References:**

Danabasoglu, G., & Gent, P. R. (2009). Equilibrium climate sensitivity: Is it accurate to use a slab ocean model? *Journal of Climate*, *22*(9), 2494–2499. https://doi.org/10.1175/2008JCLI2596.1

Ferrari, R., Jansen, M. F., Adkins, J. F., Burke, A., Stewart, A. L., & Thompson, A. F. (2014). Antarctic sea ice control on ocean circulation in present and glacial climates. *Proceedings of the National Academy of Sciences*, *111*(24), 8753 LP – 8758. https://doi.org/10.1073/pnas.1323922111

Kutzbach, J. E., He, F., Vavrus, S. J., & Ruddiman, W. F. (2013). The dependence of equilibrium climate sensitivity on climate state: Applications to studies of climates colder than present. *Geophysical Research Letters*, *40*(14), 3721–3726. https://doi.org/10.1002/grl.50724

Pincus, R., Forster, P. M., & Stevens, B. (2016). The Radiative Forcing Model Intercomparison Project (RFMIP): experimental protocol for CMIP6. *Geosci. Model Dev.*, *9*(9), 3447–3460. https://doi.org/10.5194/gmd-9-3447-2016

Shin, S.-I., Liu, Z., Otto-Bliesner, B. L., Kutzbach, J. E., & Vavrus, S. J. (2003). Southern Ocean sea-ice control of the glacial North Atlantic thermohaline circulation. *Geophysical Research Letters*, *30*(2), 68–71. https://doi.org/10.1029/2002GL015513

---

## Author Comment (AC2) · 1 Dec 2020

One point which I forgot in my review so far, but which might also get improved by more details: The efficacy of GHG @ LGM was 0.9 here. I am wondering if the full impact of CH4 on radiative forcing includes indirect effects of CH4 on stratospheric H2O and tropospheric O3. They have been proposed in Hansen et al., (2005) and they led to an additional change of 40% in radiative forcing of CH4 (Hansen et al., 2008).

**Reply:** Thanks for the interesting point and references. Our simulations used the Community Atmosphere Model version 5, which is a low-top model with prescribed stratospheric chemical tracers, such as $CH_4$, $H_2O$, and $O_3$.

We will add the following discussion in the revised manuscript: "We note that the LGM GHG forcing and efficacy in this study is calculated using a "low-top" atmosphere model with prescribed stratospheric chemical tracers and excludes indirect effects from stratosphere chemistry (Hansen et al., 2005)."

Hansen, J., Sato, M., Ruedy, R., Nazarenko, L., Lacis, A., Schmidt, G.A., Russell, G., Aleinov, I., Bauer, M., Bauer, S., Bell, N., Cairns, B., Canuto, V., Chandler, M., Cheng, Y., Genio, A.D., Faluvegi, G., Fleming, E., Friend, A., Hall, T., Jackman, C., Kelley, M., Kiang, N., Koch, D., Lean, J., Lerner, J., Lo, K., Menon, S., Miller, R., Minnis, P., Novakov, T., Oinas, V., Perlwitz, J., Perlwitz, J., Rind, D., Romanou, A., Shindell, D., Stone, P., Sun, S., Tausnev, N., Thresher, D., Wielicki, B., Wong, T., Yao, M., Zhang, S., 2005. Efficacy of climate forcings. Journal of Geophysical Research 110, D18104. doi:10.1029/2005JD005776.

Hansen, J., Sato, M., Kharecha, P., Beerling, D., Berner, R., Masson-Delmotte, V., Pagani, M., Raymo, M., Royer, D.L., Zachos, J.C., 2008. Target atmospheric CO2: where should humanity aim? The Open Atmospheric Science Journal 2, 217–231. doi:10.2174/1874282300802010217.

---

## Author Comment (AC3) · 2 Dec 2020

The authors present climate model simulations and analysis regarding climate sensitivity, radiative forcing and feedbacks in the LGM. Simulations with fixed SSTs and a slab ocean model and a model with a fully dynamic ocean component are used to analyze forcing from greenhouse gases and ice sheets. Differences between the fixed SST and slab ocean model results give "effective radiative forcings" and "efficacy" of forcings. The method is quite complicated (at least for a person not intimately familiar with these concepts). This also makes for difficult reading. I find it a little confusing that some of the so called forcings include what would be traditionally considered feedbacks, e.g. the longwave response to temperature changes illustrated in Fig. 2 would have traditionally been considered a feedback. But I guess this is the purpose, to separate effects beyond the traditional forcing/feedbacks concept. Despite the difficulty for a general reader I think technical papers like this are important because they advance the science by adding more detailed information. For this reason, I support the publication of this paper, perhaps in a slightly modified form.

**Reply:** We thank Reviewer 2 for the comments. We agree that part of the manuscript is technical, which, we think, should not be shortened, to ensure the reproducibility of our work. Importantly, to our knowledge, our manuscript is the first to adopt the new forcing-feedback framework of Sherwood et al. (2015) in a paleoclimate study, which is another reason we need to document the methodology in detail.

We emphasize that the methodology described in our study has advantages over the traditional methods in the quantification of LGM forcing and feedbacks. First, the use of the new framework (with the concept of effective radiative forcing and adjustments) fits better with the concept of climate forcing and response, i.e., the global radiative response being a linear function of surface temperature change and the forcing being the radiative forcing without any change in global mean surface temperature. Second, our method provides an effective way to quantify the magnitude and effectiveness of the ice-sheet forcing, which are important for a complete understanding of the LGM climate and the constraint on ECS.

An important conclusion is the ocean dynamical feedback, which the authors suggest increases climate sensitivity. This is a pure model analysis without consideration of observations. This is fine, but I think the paper could benefit from some additional discussion regarding observations.

E.g. the shortwave forcing from ice sheets certainly depends on the cloud cover simulated in the preindustrial (PI) model over regions that become ice covered in the LGM. E.g. if cloud cover and thus planetary albedo is high already at PI it won't change as much by adding ice compared with a low cloud cover/albedo case. I think this is a major source of uncertainty in determining the LGM shortwave forcing from ice sheets. It could be addressed by comparing model's cloud cover/albedo with present day observations in those regions. So, I suggest the authors to think about this and perhaps include a discussion about it in a revised version. If possible, a model data comparison would be useful, but it is not necessary. (My philosophy of reviewing papers is that the reviewer should not demand additional evidence, but evaluate the evidence presented.)

**Reply:** We agree with the reviewer and will add a short discussion regarding model-data comparison on the ocean circulations in the revised manuscript.

As for the cloud cover in PI simulations, we note that the existence of LGM ice sheets will make the clouds aloft thinner by occupying the space that would otherwise be taken by air masses and clouds. In other words, the cloud masking effect will be smaller at the LGM than expected from PI conditions, because clouds above ice sheets are thinner due to the existence of ice sheets.

Satellite observation shows a planetary albedo of 0.3–0.4 over the North America (see the SERES images here: https://ceres.larc.nasa.gov/resources/images/). CESM1 well-reproduces the observation with a bias less than 0.05. Changes in planetary albedo over the North America caused by the LGM ice sheets are larger than 0.3 in the model simulation. This result indicates that LGM ice sheets still provide a dominant impact on planetary albedo.

Another more major issue is the effect of sea level drop. It is not mentioned in the paper, but I think it should be, because the topographic effect of adding ice to the continents is accompanied by the effect of lower sea levels. Was this considered in the simulations presented here? In other words, was atmospheric mass conserved?

**Reply:** The effect of sea level drop has been included in our LGM simulation and the quantification of ice-sheet forcing/efficacy. The atmospheric mass was conserved. We will clarify these points in the revised manuscript.

Another issue is the vegetation response, which apparently is fixed at present day. This will presumably affect land temperatures and thus the calculations presented in the paper. A discussion is warranted. E.g. it is well known that during the LGM much of the present day boreal forest was converted to tundra. This would affect the surface albedo not only directly but also by modifying the effect of snow on the surface albedo.

**Reply:** The definition of Charney Sensitivity does not include vegetation changes, effect from which is considered as climate forcing by PALAEOSENS-Project Members (2012). The benefit of fixed vegetation in our simulations is that we can focus on the classical forcing-response processes (including clouds, water vapor, lapse rate, and snow/sea ice) within the framework of Charney Sensitivity. In the present manuscript, we point out that, even without the consideration of vegetation changes, the forcing-response processes within the atmosphere-ocean-sea ice coupled system is complicated enough that caution should be exercised when directly estimating ECS using paleoclimate data.

We will add a short discussion in the revised manuscript pointing out that forcing-feedback processes associated with the LGM vegetation and aerosol is less understood and worth further study.

Minor comments: numbers indicate line numbers

55: consider including Broccoli 2000 J. Clim. 13, 951pp and Schmittner et al., 2011 doi: 10.1126/science.1203513

**Reply:** Thanks. These will be added.

102-103: The authors compare their model with new results from Tierney et al., which is fine, but I think they should also compare to previous results e.g. the MARGO SST compilation. Also, I think there is some circularity here because I think Tierney et al. used the CESM model data for their temperature estimates, so they are not independent. I'd also suggest comparing their model's whole ocean temperature change with data from Bereiter et al., (2018, Nature 553, 39–44). It may also be useful to show or note the trend in whole ocean T if not in equilibrium.

**Reply:** Tierney et al. (2020) provided two estimates: a data assimilation result of –6.1°C (–6.5 to –5.7) and a data-only result of –5.6°C (–6.8 to –4.4). We used the data-only result for the model-data comparison to avoid any circularity.

The global volume-mean ocean temperature in the LGM simulation decreased by 0.15°C during the last 900 years. We will add this information in the revised manuscript. We do not think the comparison of the modeled whole ocean temperature with Bereiter et al. (2018) offer new insights regarding our findings, as our study uses CESM1 in a "perfect model" scenario to explore the assumptions associated with estimating ECS from knowledge of paleoclimate forcing and response.

105: Sherwood et al. should be (2015) I think

**Reply:** Yes, this will be corrected.

111: "active land model" not quite correct if vegetation is fixed

**Reply:** The "active land model" means that the land surface temperature is free to evolve, unlike the fixed SST. We will clarify this in the revised manuscript.

127: is the correction applied locally or globally?

**Reply:** The correction is applied locally. The global mean of ERF is independent on whether the correction is applied locally or globally.

225: "over high-latitude regions" Looking at Fig. 3c,f, I don't see large changes at high latitudes. I see more negative values for lambda over the western tropical Pacific and more positive values over the eastern subtropical Pacific. Would be useful to elaborate on this more.

**Reply:** Thanks for pointing this out. To better show the state dependence of the cloud feedback, we will add a plot of the zonal mean value.

234: "0.31" shouldn't this be 0.1? Does it refer to the CLD column in SOM_GHG?

**Reply:** The value should be 0.21 and it refer to the CLD column in SOM_2CO2. We will correct this.

252: typo: the "importance"

**Reply:** It will be corrected.

296: consider adding Schmittner 2003 EPSL doi:10.1016/S0012-821X(03)00291-7

**Reply:** It will be added.

311: add "in the model" to clarify that this statement refers to the model, not the real world

**Reply:** Thanks. It will be added.

362: "high latitudes" see my comment above

**Reply:** We will add a plot of the zonal mean to better show the differences over high latitudes.

---

## Author Comment (AC4) · 2 Dec 2020

We thank the Editor for careful reading of our manuscript and the helpful suggestions. We have addressed all the comments in the revised manuscript.

1. The concept of equilibrium climate sensitivity (ECS) is defined very clearly the introduction; however, I wonder if (for the benefit of some readers who may be more exclusively familiar with the palaeoclimate context) it might be useful to disambiguate 'ECS' from the term 'ES', designating 'Earth System Sensitivity', which has been introduced in the literature (e.g. by Schneider et al., Lund et al.) and which aims to include the effects of slower feedbacks in the climate system. This is just a suggestion of course; the definition that is provided for ECS per se is very clear.

**Reply:** We agree that it is better to compare ECS with ESS, especially in the paleoclimate context. We will add the following sentence in the revised manuscript.

"By tradition and for practical reasons, ECS does not account for slow feedback processes, such as changes in vegetation, cryosphere and ocean circulation, effects of which has been included in the Earth system sensitivity (e.g. Lunt et al., 2010)."

2. Line 46: reference is made to the term 'efficacy' here, and elsewhere in the introductory text; however, it is only clearly defined on line 162 and in equation 4. I would suggest that a brief verbal definition of the term be provided up front (e.g. along the lines of "the ratio of the climate sensitivity parameter (K/wm-2) derived for a given forcing relative to that for a doubling of atmospheric CO2", or "the ratio of the warming effect attributed to a given forcing, relative to that due to a doubling of atmospheric CO2 under pre-industrial conditions".. etc. . .). I realise that it might be hard to be succinct and accurate at the same time, but this is an important concept in the paper and it will be important to make it clear to readers early on.

**Reply:** Thank you for the suggestion. We will add a verbal definition of efficacy in the revised manuscript.

3. Line 73: I found this sentence hard to decipher, and wondered if the following was an accurate reflection of what was intended: ". . .to provide a complete quantification of the LGM LIS and GHG forcing, and their respective 'efficacies', using a suite of climate simulations."

**Reply:** Thank you for the suggestion. We will change the revised manuscript accordingly.

4. Line 117: It is not clear what the last sentence means to say; please rephrase to clarify (e.g. do adjustments reflect changes that occur 'as a direct result of a given forcing, without mediation by global average temperature change, i.e. not including the Planck feedback?'). It is hard to see immediately what changes in temperature, clouds etc. . ., would be mediated by 'global average' temperature change specifically, as opposed to local/regional changes, apart from the Planck feedback on global longwave output. For example, I understand that sea-ice and snow cover changes that arise from a cooling caused by a GHG change would be excluded as 'adjustments', but do these really arise from 'global average' temperature changes?

**Reply:** The concept of effective radiative forcing and adjustments is developed to quantify the forcing such that the forcing excludes radiative effects associated with surface temperature change (forcing being independent of response) and that the top-of-atmosphere energy imbalance is closer linked to global mean surface temperature. The concept fits better with the forcing-feedback framework.

In the revised manuscript, we will add citation to Figure 1 of Sherwood et al. (2015) to clarify the concept.

5. Line 160: Is it possible to clarify this sentence? E.g. ". . .represents the global surface air T change associated with an effective radiative forcing, but that is driven indirectly (by SST change)"? Is my suggestion accurate?

**Reply:** We will re-write the sentence as: "$\Delta T_{SOM} - \Delta T_{fsst}$ represents the SST-mediated surface air temperature changes that is *driven* by $ERF_{fsst}$".

6. Line 214: As I will expand upon a little more below, I find the phrasing

'overestimation/underestimation' somewhat misleading at times, or at least open to misunderstanding. For example, here, I would suggest that it might be clearer to state something like: ". . .this APRP approach overestimates the shortwave radiative forcing that is attributable exclusively to changes in LIS extent, as it includes the radiative effect of snow increases over ice sheets (or regions with shelf exposure); the albedo of fresh snow is considerably larger ".

**Reply:** Thank you for the suggestion. We will make the suggest change.

7. Line 216: Similarly I would suggest a minor clarification such as: "The snow-induced overestimation [of the LIS direct contribution] is larger if the cooling over ice sheets is greater."

**Reply:** Thank you for the suggestion. We will make the suggest change.

8. Line 218: I think that the use of plural for simulations might be better, i.e.: " is greater in coupled simulations. . . atmosphere-only simulations. . .

**Reply:** Thank you. We will make the suggested change.

9. Line 224: Is the study of M. Crucifix (2006, Does the Last Glacial Maximum constrain climate sensitivity?, Geophys. Res. Lett., 33, L18701) relevant here at all (with respect to the temperature dependence of cloud feedbacks)?

**Reply:** Thank you. Crucifix (2006) is relevant and will be added.

10. Line 258: ". . .the importance of using. . .". I would also suggest adding for clarity: " using efficacy to evaluate the overall effectiveness of their radiative forcing as compared to a doubling of atmospheric CO2."

**Reply:** Thank you for the suggestion. We will make the suggested change.

11. Section 3.3: The point here seems to be that the system is broadly linear (at least by virtue of any regional non-linearities cancelling out globally perhaps?); however, I wondered if it would it be justified in your view to add a caveat that this point applies primarily to an evaluation of short-term impacts (i.e. from fast feedbacks)?

**Reply:** Yes, you are right. The last sentence of Section 3.3 is a caveat that the linearity only holds for the global mean and is not true for regional forcing and response. We will extend the caveat to suggest that the linearity dose not necessarily hold for long-term feedback processes.

12. Line 273: Surely ocean interior temperatures will not be in equilibrium after 60 years, or if they are in the SOM some caution is warranted in extrapolating to the real global ocean? I simply invite your consideration of whether any clarification is needed here.

**Reply:** SOM simulations usually reach equilibrium in less than 50 years (Danabasoglu & Gent, J. Climate, 22(9), 2494–2499, 2009). SOM simulations are not meant to approximate the real global ocean but used to explore response of the climate system without the consideration of ocean dynamics. Comparing SOM against the fully coupled simulations enables us to quantify the role of ocean dynamics on surface temperature.

13. Line 289: Would it be more complete to state that the remote impact on the SO reflects the impact on SO stratification of a displacement in tropical atmospheric circulations, etc..?

**Reply:** We meant the other way around, i.e., SO cooling impacts the subtropics through shifting the ITCZ and trade winds.

We will re-write the sentence as: "This reflects a remote impact of the SO processes on the lower latitudes through changing tropical atmospheric circulations."

14. Line 333: Again, can I suggest to add the clarification: "We note that, due to the inclusion of snow effects in the forcing quantification, the APRP-based approach overestimates the direct shortwave albedo effects that are attributable only to changing LIS extent"? My point is that it is only an 'overestimation' if one wants to strip out the knock-on effects of a changing LIS, to consider only

direct impacts. Otherwise, one could argue, conversely that the 'real' impact of changing LIS extent is actually underestimated by an approach that does not consider the knock-on effects.

**Reply:** We agree and will make changes as you suggested.

15. Line 352: Here again I would suggest to alter slightly the language used, for clarity. E.g. "If we do not remove the ocean dynamical feedbacks. . .".

**Reply:** We agree and will make changes as you suggested.

16. Line 357: Similarly, can I suggest for your consideration: "In sum, this exercise highlights the importance of the ocean dynamical feedback, which, if included, may cause an overestimation of the ('fast feedback') ECS value using reconstructions of LGM forcings/responses." To my mind, 'neglecting' the ocean dynamical feedback would be the same as not stripping it out of the radiative/temperature effects, which is somewhat confusing.

**Reply:** Thanks. We will change the sentence to read: "In sum, this exercise highlights the importance of the ocean dynamical feedback, which, if not accounted for, may cause an overestimation of the ('fast feedback') ECS value using reconstructions of LGM forcings/responses."

17. Line 375: In the same vein as the above comments, can I propose for you to consider: "LGM-based ECS calculations that neglect to remove ocean dynamical effects produce an overestimation [of fast feedbacks/sub-centennial impacts] by approximately 25%." My point is simply that I it may be important to make sure no one misunderstand this statement as suggesting that the ocean dynamical feedback dampens warming, when in fact it amplifies it.

**Reply:** Thank you. We will change the sentence to read: "LGM-based ECS calculations that fail to account for this ocean dynamical effects produce an overestimation of fast feedbacks by approximately 25%."

18. Finally, I can't help but add to Referee 1's comment number 19, that radiocarbon evidence from the LGM is likely more useful as a constraint on large-scale mixing/air-sea exchange of heat/carbon than is d13C, which notably has a non-conservative component due to biological export production. In any event, both lines of evidence would indeed suggest greater stratification/sea ice coverage, not less.

**Reply:** Thank you. Please see also our response to Review 1's comments. In the manuscript, we meant to describe the upper-ocean stratification, which is most relevant to the mixed layer heat budget and the interaction with sea ice.

Major features of deep-ocean circulation and seawater characteristics in our CESM1 simulations agree well with findings from proxy reconstructions, including an expansion of the Antarctic Bottom Water, a shallower North Atlantic Deepwater, an increase in abyssal stratification, and a saltier and colder southern-source deep water. We will add a short discussion on model-data comparison of the LGM ocean circulation.

---

## Author Response (AR1)

Dear Dr. Skinner,

Thank you for carefully assessing our manuscript and your constructive suggestions. We agree with you that these are important points that need to be clarified. Please find below our response.

When you prepare your revised manuscript, I would like to encourage you to reconsider your dismissal of one particular review comment, regarding the illustration of the 'LGM' CESM ocean state (reviewer 2). You state that:

"The global volume-mean ocean temperature in the LGM simulation decreased by 0.15°C during the last 900 years. We will add this information in the revised manuscript. We do not think the comparison of the modeled whole ocean temperature with Bereiter et al. (2018) offer new insights regarding our findings, as our study uses CESM1 in a "perfect model" scenario to explore the assumptions associated with estimating ECS from knowledge of paleoclimate forcing and response."

However, it seems entirely relevant to me (and I invite you to respond) whether or not the ocean is at thermal equilibrium in your LGM simulation, and whether or not its state is one in which a significant net heat flux to/from the ocean exists, due to e.g. disequilibrium effects. A global ocean cooling of only 0.15°C is quite far from a realistic equilibrium LGM ocean state, based on observations such as the noble gas isotope measurements of Bereiter et al. (2018) for example. I would therefore suggest that when noting the mean ocean temperature change of 0.15°C in your LGM simulation (which i agree is important) you also provide a short discussion of why this result differs from observations such as those of Bereiter et al. (2018) (i.e. is it due to a disequilibrium ocean state?), and also what arguments you can advance for why your analysis and conclusions regarding ocean dynamical feedbacks etc... are unaffected by the ocean's energy/heat budget not being at equilibrium. In response to my own comments on a related theme you have proposed that the surface ocean is at equilibrium, even if the deep ocean is indeed far from equilibrium. I still think it is important to comment on this issue in the manuscript, since it is not obvious that the surface ocean or atmosphere can really be at thermal equilibrium if the deep ocean is in the process of accumulating heat over thousands of years (with a fixed TOA energy balance). The resulting changes might be slow, but they will accumulate into a different 'true LGM' state. Or, to state things differently, if you are indeed analysing a disequilibrium ocean state, then it might be more accurate to see the simulation as a transient cooling experiment and not as a comparison with the 'LGM', as modelled or reconstructed in other studies. Please therefore add a short note on why you believe it is not relevant for your study that the deep ocean heat budget is not at equilibrium in your simulations (if indeed this is the case).

**Reply:** We point out that our findings on the effective radiative forcings of LGM GHG and LIS and their efficacy are independent on the equilibrium state of the fully coupled LGM simulation. LGM forcing and efficacy are obtained using fixed-SST and slab ocean simulations with a preindustrial SST/sea ice and ocean dynamical effects.

We agree with you that the trend in the whole-ocean temperature could have a small impact on the magnitude of the ocean dynamical effect. We have now acknowledged this by stating: "Due to limited computing resources, our fully coupled LGM simulation has a cooling trend in the deep ocean (see Section 2.1), which will not impact our results on LGM radiative forcing/efficacy but will likely cause an underestimation of LGM ΔGMST and the contribution of the ocean dynamical

feedback in the model." Please see Lines 123–124 and 444–447 in the revised manuscript with tracked changes.